# Repurposing the orphan drug nitisinone to control the transmission of African trypanosomiasis

**Marcos Sterkel**[1]☯*, **Lee R. Haines**[2]☯, **Aitor Casas-Sánchez**[2], **Vincent Owino Adung'a**[3,4], **Raquel J. Vionette-Amaral**[2], **Shannon Quek**[5], **Clair Rose**[2], **Mariana Silva dos Santos**[6], **Natalia García Escude**[2], **Hanafy M. Ismail**[2], **Mark I. Paine**[2], **Seth M. Barribeau**[7], **Simon Wagstaff**[5], **James I. MacRae**[6], **Daniel Masiga**[3], **Laith Yakob**[8]*, **Pedro L. Oliveira**[9,10], **Álvaro Acosta-Serrano**[2,5]*

**1** Centro Regional de Estudios Genómicos, Facultad de Ciencias Exactas, Universidad Nacional de La Plata, Argentina, **2** Department of Vector Biology, Liverpool School of Tropical Medicine, United Kingdom, **3** International Centre of Insect Physiology and Ecology, Nairobi, Kenya, **4** Department of Biochemistry and Molecular Biology, Egerton University, Kenya, **5** Department of Tropical Disease Biology, Liverpool School of Tropical Medicine, United Kingdom, **6** Crick Institute, London, United Kingdom, **7** Department of Ecology Evolution & Behaviour, Institute of Integrative Biology, University of Liverpool, United Kingdom, **8** Department of Disease Control, London School of Hygiene and Tropical Medicine, United Kingdom, **9** Instituto de Bioquímica Médica Leopoldo de Meis, Universidade Federal do Rio de Janeiro, Rio de Janeiro, Brazil, **10** Instituto Nacional de Ciência e Tecnologia em Entomologia Molecular (INCT-EM), Rio de Janeiro, Brazil

☯ These authors contributed equally to this work.
* msterkel@conicet.gov.ar (MS); laith.yakob@lshtm.ac.uk (LY); alvaro.acosta-serrano@lstmed.ac.uk (AAS)

**Data Availability Statement:** All relevant data are within the paper and its Supporting Information files, except for the metabolomics dataset, which has been deposited into the MetaboLights

## Abstract

Tsetse transmit African trypanosomiasis, which is a disease fatal to both humans and animals. A vaccine to protect against this disease does not exist so transmission control relies on eliminating tsetse populations. Although neurotoxic insecticides are the gold standard for insect control, they negatively impact the environment and reduce populations of insect pollinator species. Here we present a promising, environment-friendly alternative to current insecticides that targets the insect tyrosine metabolism pathway. A bloodmeal contains high levels of tyrosine, which is toxic to haematophagous insects if it is not degraded and eliminated. RNA interference (RNAi) of either the first two enzymes in the tyrosine degradation pathway (tyrosine aminotransferase (TAT) and 4-hydroxyphenylpyruvate dioxygenase (HPPD)) was lethal to tsetse. Furthermore, nitisinone (NTBC), an FDA-approved tyrosine catabolism inhibitor, killed tsetse regardless if the drug was orally or topically applied. However, oral administration of NTBC to bumblebees did not affect their survival. Using a novel mathematical model, we show that NTBC could reduce the transmission of African trypanosomiasis in sub-Saharan Africa, thus accelerating current disease elimination programmes.

repository (https://www.ebi.ac.uk/metabolights/) under accession number MTBLS2166.

**Funding:** This work was supported by UK Medical Research Council (MRC) Confidence in Concept awards 2016-17 MC_PC_16052 and 2017-18 MC_PC_17167 to AA-S and MIP (https://mrc.ukri.org/), and the Biotechnology and Biological Sciences Research Council (BBSRC) Anti-VeC award AV/PP0021/1 to AA-S and LHR (https://bbsrc.ukri.org). MS was supported by grants from FONCyT PICT 2017-1015 (http://www.foncyt.mincyt.gov.ar/), BBSRC Anti-VeC AV/TTKE/0011 and CAPES/FAPERJ No E-26/102.837/2011 (http://www.faperj.br/). PLO was supported by FAPERJ E-26/210.246/2018, CAPES 9152899772/CAPES-PRINT760102P (https://www.gov.br/capes/pt-br) and CNPq 09078/2018-2 (www.http://www.cnpq.br). SW was supported by LSTM Research Computing Unit. JIM and MSS were supported by The Francis Crick Institute which receives its core funding from Cancer Research UK FC001999 (https://www.cancerresearchuk.org), the MRC FC001999, and the Wellcome Trust FC001999 (https://wellcome.org/). Confocal images were supported by a Wellcome Trust Multi-User Equipment grant 104936/Z/14/Z. The funders had no role in study design, data collection and analysis, decision to publish, or preparation of the manuscript.

**Competing interests:** The authors have declared that no competing interests exist

**Abbreviations:** AAT, animal African trypanosomiasis; ARCU, Animal Rearing Unit and Containment Unit; ATSB, attractive targeted sugar bait; BSA, bovine serum albumin; dsRNA, double-stranded RNA; FAH, fumarylacetoacetase; GFP, green fluorescence protein; HAT, human African trypanosomiasis; HB, homogenization buffer; HCD, high-energy collisional dissociation; HIS1, HPPD Inhibitor Sensitive 1; HPLA, 4-hydroxyphenyl lactic acid; HPLC, High Performance Liquid Chromatography; HPPD, 4-hydroxyphenylpyruvate dioxygenase; HPPA, 4-hydroxyphenylpyruvate; HT-1, hypertyrosinemia type I; IACUC, Institutional Animal Care and Use Committee; *icipe*, International Centre of Insect Physiology and Ecology; LC-MS/MS, liquid chromatography-tandem mass spectrometry; MDA, mass drug administration; NFW, nuclease-free water; NTBC, nitisinone; PBQC, pooled biological quality control; PRM, parallel reaction monitoring; qPCR, quantitative polymerase chain reaction; RFU, relative fluorescence unit; RNAi, RNA interference; RT-PCR, reverse transcription PCR; TAT, tyrosine aminotransferase.

## Introduction

Human African trypanosomiasis (HAT), also known as sleeping sickness, is a parasitic disease caused predominantly by the parasite *Trypanosoma brucei gambiense*. These parasites are transmitted to a vertebrate host when infected tsetse flies (*Glossina* spp.) blood feed. HAT currently affects 3,500 people/year; most patients live in the Democratic Republic of the Congo and an estimated 70 million people remain at risk of infection in sub-Saharan Africa [1]. Tsetse also spread animal African trypanosomiasis (AAT), which causes high mortality rates in livestock and consequently severely limits animal production [2]. As no vaccine for either HAT or AAT exist, and drug treatments are often difficult to obtain, tsetse population control remains essential to limit the spread of trypanosomiasis. In the last decades, tsetse control tools such as aerial spraying of insecticides (pyrethroids), visual- and odour-baited tsetse traps, insecticide-treated livestock, live traps, insecticide-impregnated traps and targets, and sterile male releases have been employed [3–7]. Despite such efforts, because AAT and HAT persist in these endemic areas, both economic development and public health continue to be jeopardised [8]. Consequently, a novel complementary strategy to control these parasitic diseases is highly desired.

Tsetse, like other blood-feeding arthropods, ingest large quantities of blood and often exceed twice their body weight in a single meal [9]. Since more than 85% of blood dry weight consists of proteins, large quantities of amino acids are released in the midgut during blood-meal digestion [10]. Previously, we showed that blocking tyrosine catabolism after a bloodmeal is lethal in mosquitoes, ticks, and kissing bugs due to the accumulation of toxic quantities of tyrosine [11]. However, inhibiting tyrosine catabolism in non-blood-feeding insects is harmless, which further provides evidence for the essentiality of this pathway for haematophagy [11]. In the present work, we evaluated how tsetse physiology was controlled by two enzymes in the tyrosine catabolism pathway: tyrosine aminotransferase (TAT) and 4-hydroxyphenyl-pyruvate dioxygenase (HPPD). The drug nitisinone (2-(2-nitro-4-trifluoromethylbenzoyl)-1,3cyclohexanedione; NTBC), also known as Orfadin, is an HPPD inhibitor currently used to treat patients with the genetic disease hypertyrosinemia type I (HT-1) [12], and is under clinical evaluation for the treatment of alkaptonuria [13]. NTBC was lethal to blood-fed tsetse flies. NTBC treatment, either administered orally as an endectocide or topically to the insect cuticle, causes the accumulation of tyrosine and 4-hydroxyphenyl lactic acid (HPLA) metabolites, which leads to initial fly paralysis followed by tissue destruction within 18 hours of the blood-meal. Our results provide evidence that NTBC could be used as an eco-friendly synergistic strategy alongside current tsetse control practices.

## Results

### Tyrosine detoxification is essential in tsetse

Tyrosine catabolism is a highly conserved pathway (Fig 1A) in most eukaryote and prokaryote species with only a few exceptions such as the pea aphid, *Acyrthosiphon pisum* [14], and trypanosome parasites [15]. The genes encoding TAT and HPPD proteins were identified in five *Glossina* species [16], as well as in all hematophagous arthropod species with sequenced genomes (S1 Fig). RNA interference (RNAi), of either *TAT* or *HPPD* genes, was lethal to flies once they fed on blood. However, 100% mortality was not observed, which may be explained by incomplete knockdown (Fig 1B and S2 Fig). This lethality was further validated by feeding flies with blood supplemented with mesotrione, an HPPD inhibitor widely used as a selective herbicide on corn crops under the brand name *Callistro*, Syngenta (S3A Fig). The mesotrione concentration that killed 50% of the insects 24 hours after administration (LC$_{50}$) was 357.7 μM

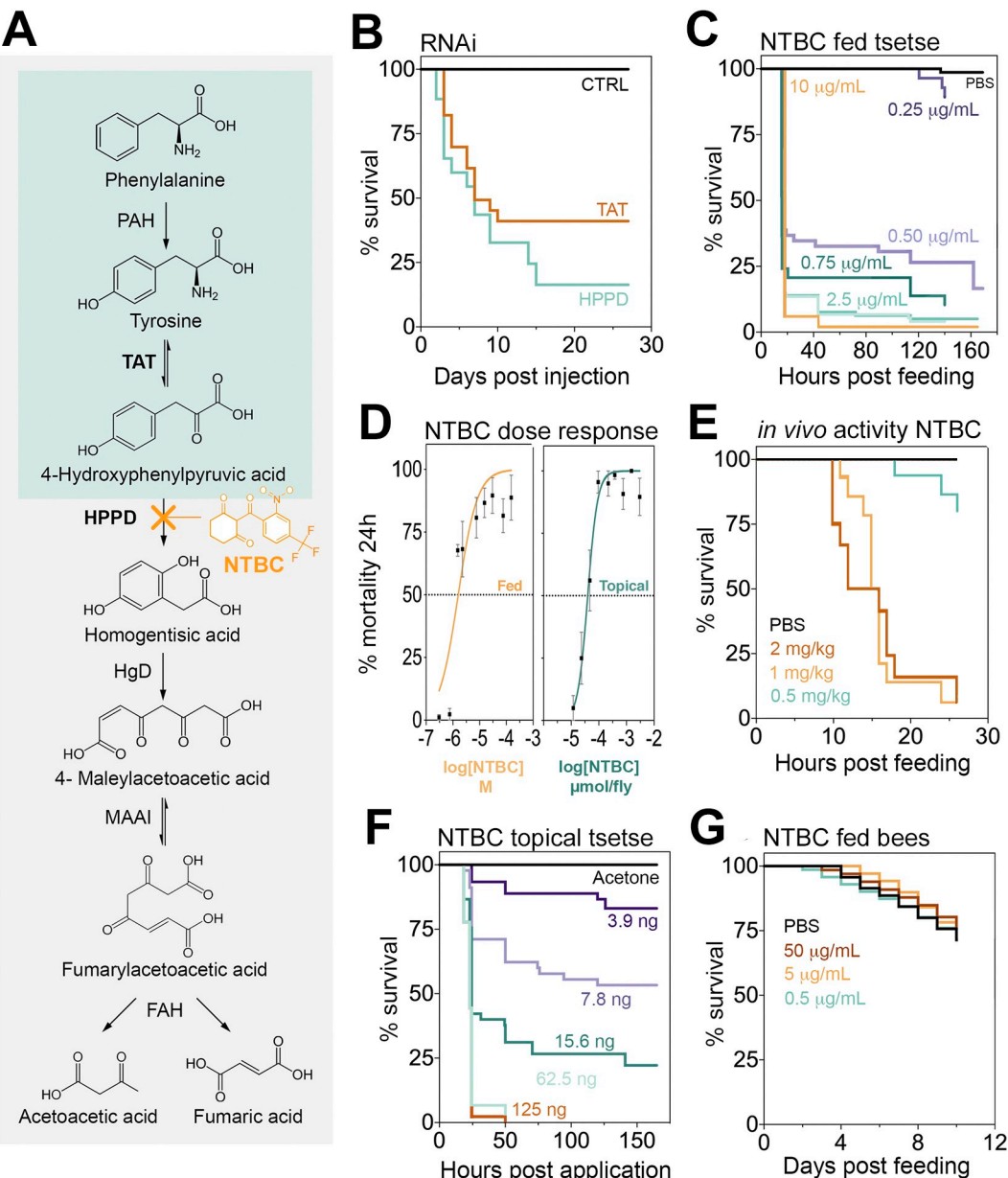

**Fig 1. Inhibiting tyrosine catabolism is lethal for tsetse, but not for bees.** (A) Tyrosine catabolism pathway. (B) Survival of *Glossina morsitans morsitans* when injected with dsRNA to knockdown TAT or HPPD. Two independent experiments were performed, each with *n* = 10–13, 12–16, and 12–14 for dsGFP (CTRL), dsTAT, and dsHPPD, respectively. Five insects from each group were dissected three days after a bloodmeal (PBM) to assess gene knockdown efficiency (S2 Fig). (C) Survival of *G. m. morsitans* fed with horse blood supplemented with NTBC or PBS (Control. 9:1; v/v). Three to six independent experiments were performed for different doses, each with *n* = 10–25 insects. The total number of flies (male and female) used was 914 (All doses applied are shown in S3B Fig). (D) Dose-response curves for *G. m. morsitans* survival 24 hours PBM: NTBC feeding and topical application assays (mean ± SEM). (E) Male *Glossina pallidipes* survival after feeding on PBS- or NTBC-treated rats. Doses are presented as mg of NTBC per kg of rat body weight. Three independent experiments were performed per dose, each with *n* = 12–15 flies. The total number of male flies used was 86. (F) Survival of NTBC-treated (topically) *G. m. morsitans* after a bloodmeal. Three independent experiments were performed for each dose, each with *n* = 10–20 insects. The total number of male flies used was 449. (G) Bee survival when maintained on a 50% sugar solution supplemented with NTBC (or PBS) and pollen. Two independent experiments were performed, each with *n* = 25 insects. The total number of bees used was 200. The underlying data for this figure can be found in S1 Data. CTRL, control; dsGFP, double-stranded green fluorescence protein; dsRNA, double-stranded RNA; FAH, fumarylacetoacetase; HgD, homogentisate 1,2-dioxygenase; HPPD, 4-hydroxyphenylpyruvate dioxygenase; MAAI, maleylacetoacetate isomerase; NTBC, nitisinone; PAH, phenylalanine hydroxylase; PBM, post-blood meal; TAT, tyrosine aminotransferase.

(95% CI: 222.5 to 512.4) (S3A and S3C Fig). This lethal concentration of mesotrione is approximately 30× higher than the drug concentration detected in human plasma (4 μg/ml (11.78 μM)) after volunteers received an oral dose of 4 mg/kg body weight [17]. No differences in susceptibility to mesotrione (or NTBC) were observed between fly sex (S4 Fig).

## Ingestion of NTBC is lethal to tsetse

HT-1 is a severe human genetic disease caused by a mutation in the gene encoding for the last enzyme of the tyrosine catabolism pathway, fumarylacetoacetase (FAH). This mutation causes the accumulation of toxic metabolites in blood and tissues. The only drug available to minimise the effect of HT-1 is the orphan drug NTBC. As an HPPD inhibitor, NTBC prevents the buildup of toxic products derived from fumarylacetoacetate accumulation [18]. NTBC is remarkably safe to use with few reported side effects in <1% of patients [19,20]. When NTBC was fed to tsetse flies in an artificial bloodmeal, it was approximately 173 times more potent than mesotrione with a $LC_{50}$ = 2.07 μM (95% CI: 0.709 to 4.136) (Fig 1C and 1D). This lethal concentration is approximately 12 times lower than the concentration of NTBC persisting in human plasma (8 μg/ml (24.3 μM)) after administering a standard therapeutic oral dose (1 mg/kg) [17]. The NTBC half-life in human plasma is 54 hours [17], so assuming linear drug degradation, human blood would remain toxic to tsetse for at least a week after a single therapeutic dose. These results suggest that NTBC used as endectocide (here defined as a drug or insecticide that is administered to humans or animals to control parasites) [21–23] could be used to decrease tsetse populations as part of a drug-based vector control strategy. Furthermore, it did not matter if the tsetse flies were infected with *T. brucei*; NTBC lethality remained unaltered in infected flies (S5 Fig) suggesting that, unlike proline metabolism [24], the tsetse tyrosine degradation pathway does not seem to be affected during trypanosome infection.

## Tsetse fed on NTBC-treated mice die

To evaluate the potential of using NTBC as an endectocide, the *in vivo* efficacy of NTBC was assessed. Colony-reared tsetse were allowed to feed on rats that had been orally treated with different NTBC concentrations. We observed that after 26 hours of feeding on rats receiving an oral dose of NTBC equal to 1 mg/kg (therapeutic dose recommended for humans with HT-I), approximately 90% of the flies died, as reflected in our previous data (Fig 1E). Furthermore, compared to control flies, tsetse fed on NTBC-treated rats showed the same phenotypic characteristics (including darkness of the abdomen (S6 Fig)) as those fed on NTBC-blood in artificial membrane feedings (see below). Together, these data provide direct evidence that NTBC could be used as an endectocide for tsetse control.

## NTBC ingestion causes insect paralysis and systemic tissue destruction

Treatment of tsetse with either mesotrione- or NTBC-supplemented blood presented the same unique physiological changes, which has two distinct phases we classified as "early stage" and "death" phenotype. The early stage phenotype was observed 8 to 10 hours after NTBC (or mesotrione) ingestion in the bloodmeal, i.e., the flies remained alive (evidenced by red eyes) but were unable to fly (often upside down; S1 Video, S7 Fig). Exposure to bright light or tarsal stimulation often produced only a short burst of leg waving. Small, white, rhomboidal crystals (previously identified as tyrosine crystals [11]) were observed on the outside surface of the midgut epithelium against the body wall cuticle. Also, dark brown melanin-like deposits had formed in different tissues, such as the fat body, salivary glands, and flight muscle (Fig 2A and 2B). Disorganisation of tissues in the digestive tract such as the proventriculus and midgut was

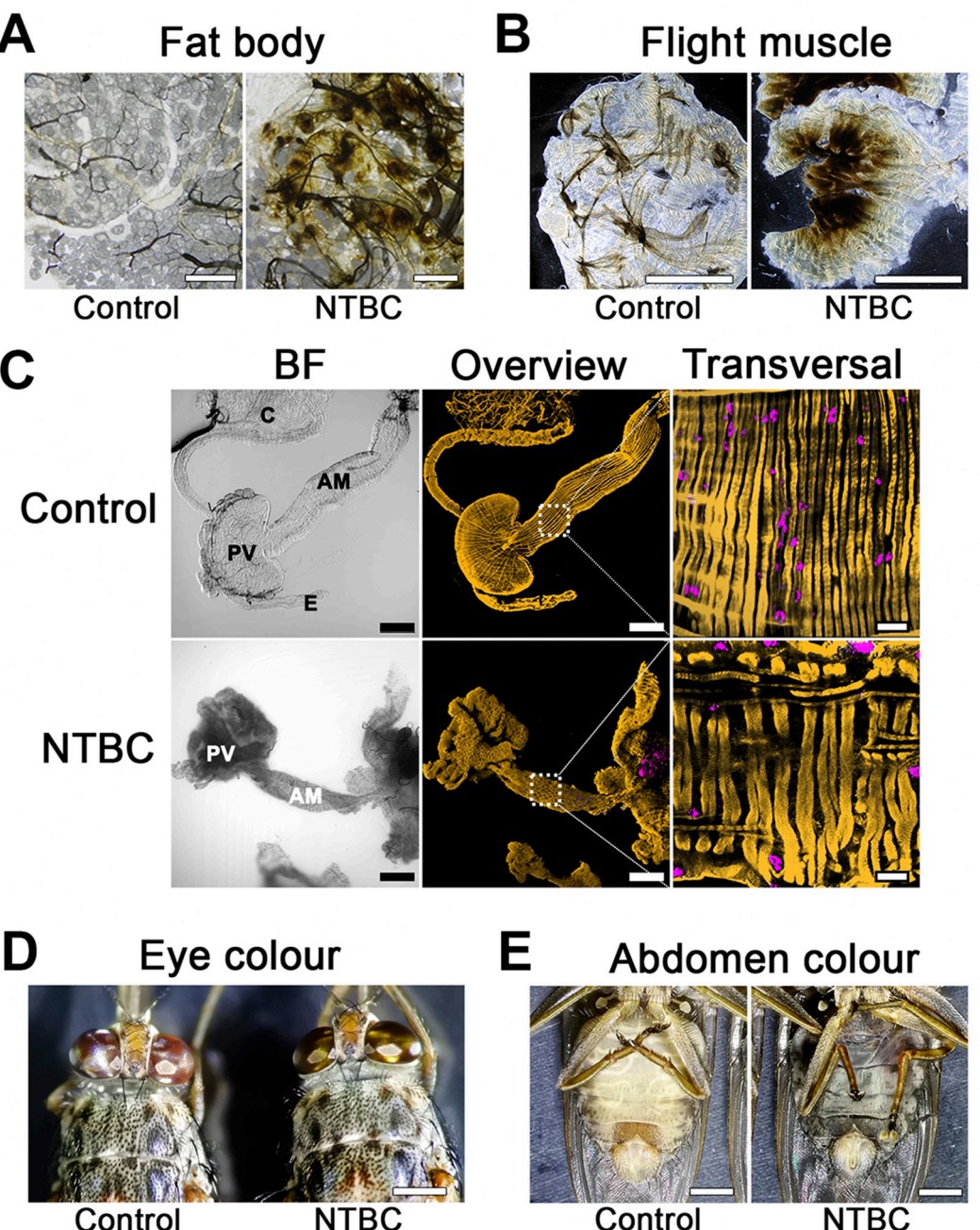

**Fig 2. Tsetse phenotypes observed upon NTBC treatment.** (A) Fat body. Scale bars = 100 μm (control) and 80 μm (treated). (B) Flight muscle. Scale bars = 1 mm. (C) Bright field (left) and fluorescence 3D-reconstructed (middle) overviews of the anterior midgut (AM), proventriculus (PV), crop (C), and oesophagus (E) from tsetse fed either without (Control) or with NTBC. Fluorescence channels merge SiR-actin staining (yellow) and DAPI (magenta). Scale bars = 200 μm (100×). Detailed sections (right) of transversal muscular fibres from the outer muscular tissues in the anterior midgut region. Scale bars = 20 μm (630×). (D) and (E) Death phenotype. Scale bars = 1 mm. Flies are already dead and the abdomens are "liquified", black and filled with undigested blood. BF, brightfield; NTBC, nitisinone.

evident (Fig 2C) and likely contributed to intestinal fragility, compromised digestion, and leaking of the midgut content into the haemocoel. Reproductive and other non-digestive abdominal tissues remained intact.

The death phenotype was observed between 15 to 18 hours after treatment and was characterised by a prolonged shift in eye colour from red to golden brown (Fig 2D). NTBC-treated tsetse showed extreme cuticular thinning and complete loss of abdominal elasticity. The abdomen was strikingly distended, dark, and filled with blood that had leaked into the haemocoel due to damage of the gut epithelium (Fig 2E). The ingested blood was blackened as though it had been oxidised. Most of the internal tissues and organs (e.g., gut, testes, ovaries, salivary glands, and Malpighian tubules) were completely destroyed and could not even be recognised in the abdomen during dissection, suggesting extensive autolysis (S8 Fig); only the hindgut, rectum, and trachea remained identifiable. Furthermore, the fat body also disappeared and only lipid droplets could be seen floating in the haemocoel. From the outside, the thorax appeared normal, but upon dissection, we observed the flight muscles had detached from the thorax and become melanised, which explains why tsetse quickly lost the ability to fly.

## NTBC leads to an accumulation of free tyrosine and 4-hydroxyphenyl lactic acid

To better understand the metabolic changes in tsetse exposed to NTBC, haemolymph from NTBC-treated (and control) flies was collected at different time points. Metabolomic analysis revealed increased levels of tyrosine and HPLA (Fig 3). The knockdown of *TAT* gene, which is expected to reduce levels of 4-hydroxyphenylpyruvate (HPPA) and, in consequence, produce less HPLA, caused the same lethal phenotype observed upon HPPD inhibition. Collectively, this suggests that tyrosine accumulation is the likely cause of the flies' death. Furthermore, feeding the flies with blood supplemented with HPLA or the injection of HPLA into the fly haemocoel did not affect their survival (S1 Table). In contrast to tyrosine and phenylalanine, the level of all other detected amino acids was reduced (Fig 3C and 3D), which may reflect a lower digestion rate in NTBC-treated flies.

## Toxicity associated to insect HPPD inhibition depends on the protein concentration in the bloodmeal and not haem

A potential cause of toxicity among tyrosine catabolism inhibition could be haem. Haem is a product of blood (haemoglobin) digestion that becomes toxic by amplifying reactive oxygen species [25]. To investigate if haem contributed to the lethal phenotype, flies were fed with horse serum (red blood cells were removed to reduce haemoglobin) supplemented with mesotrione. Fly mortality remained high even in the absence of haemoglobin, which suggested haem was not involved in the toxicity of HPPD inhibitors. Furthermore, this experiment demonstrated that the protein content in horse serum (51 to 72 mg/ml) [26] is high enough to produce toxic levels of tyrosine (Fig 4A).

In order to assess the quantity of protein necessary to cause the lethal phenotype, flies were fed with PBS or sugar supplemented with a fixed concentration of NTBC (lethal when added to bloodmeal) and different concentrations of bovine serum albumin (BSA) as the sole protein source. As expected, flies that fed on NTBC-supplemented diluent (protein-free) did not die, whereas NTBC toxicity showed a clear dependence on the protein content of the meal. The observed $LC_{50}$ was 12.5 mg/ml of protein (95% CI: 11.09 to 13.67) (Fig 4B). Only a small percentage (approximately 6%) of flies died 24 hours after feeding with either PBS-BSA (34 mg/ml) or fructose-BSA without NTBC (S9 Fig), indicating that the 24-hour mortality was due to the addition of NTBC.

## Topical application of NTBC kills tsetse

To investigate if NTBC-induced mortality was restricted to ingestion, we tested if the drug could be absorbed through the insect cuticle. Topically applying NTBC to the fly thorax caused

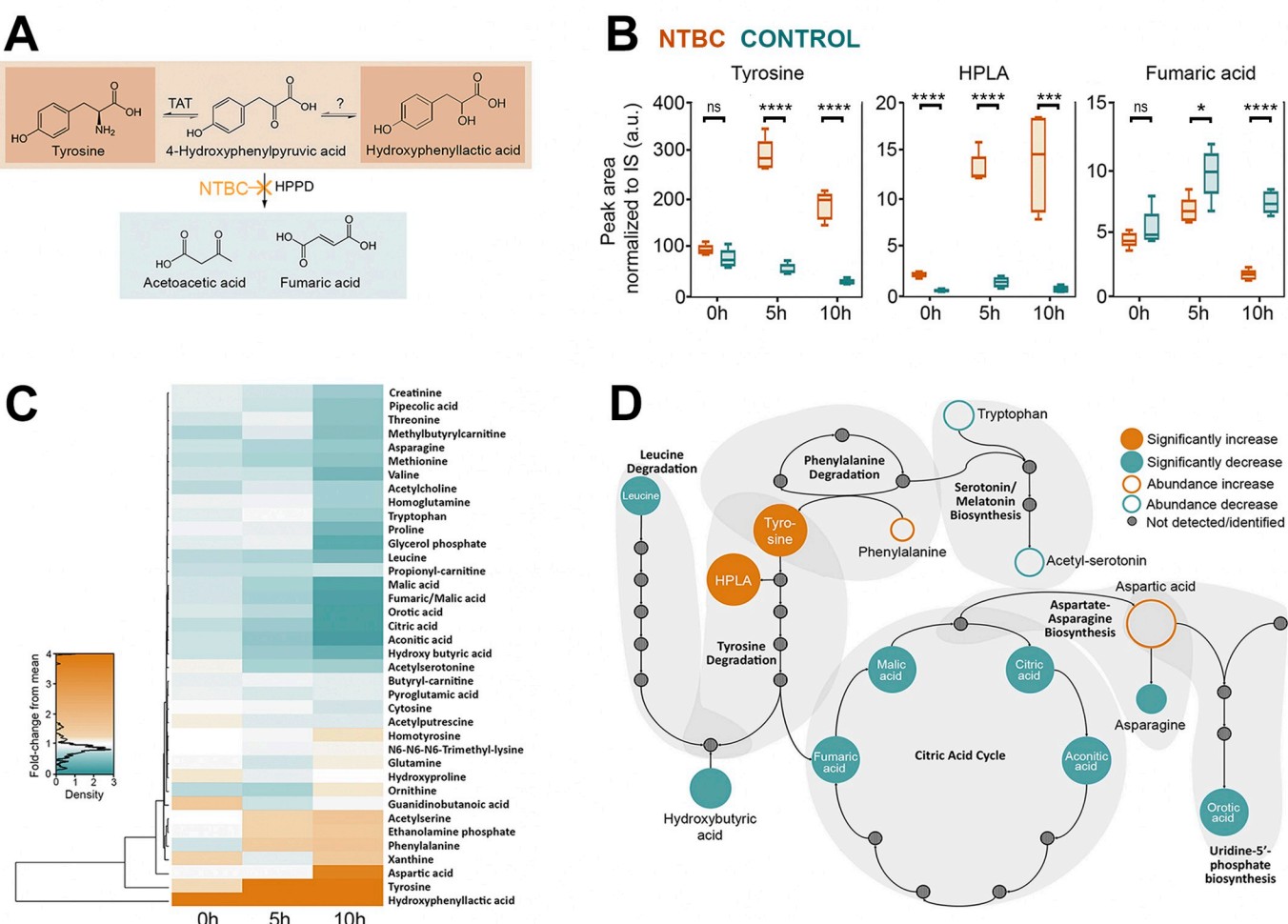

**Fig 3. Metabolomic analysis of tsetse haemolymph upon NTBC treatment at different times post-bloodmeal (0, 5, and 10 hours).** Five samples (in each group and for each time point) collected from two independent experiments were processed for and analysed by mass spectrometry. (A) Schematic representation of the tyrosine catabolic pathway highlights the main metabolites accumulated and the decreased production of final products. (B) Peak area normalised to internal standards of differentially regulated metabolites in the tyrosine catabolism pathway. ns = $P > 0.05$, * = $P \leq 0.05$, ** = $P \leq 0.01$, *** = $P \leq 0.001$, **** = $P \leq 0.0001$. (C) A heat map graphically presents the metabolites identified. (D) Schematic representation of the metabolic pathways indicates metabolites that were differentially regulated in NTBC-treated flies. The underlying data for this figure can be found in S1 Data. HPLA, 4-hydroxyphenyl lactic acid; HPPD, 4-hydroxyphenylpyruvate dioxygenase; NTBC, nitisinone; TAT, tyrosine aminotransferase.

high mortalities after tsetse took a bloodmeal, with an $LD_{50}$ of 39 picomoles/fly (95% CI: 9 to 90) (Fig 1D and 1F), 24 hours after bloodmeal ingestion. We compared this NTBC dose against a standard pyrethroid insecticide (deltamethrin) that tsetse are highly susceptible to, and the $LD_{50}$ was 0.116 picomoles/fly (95% CI: 0.093 to 0.145) (S10 Fig), which means that deltamethrin is approximately 336 times more potent than topically delivered NTBC.

In the field situation where HPPD inhibitors could be absorbed through the insect cuticle, fly contact with the drug could occur either before or after a bloodmeal. Thus, it is important to know how long NTBC activity persists in insects, particularly if tsetse feed after cuticular exposure. High fly mortality was observed even when a tsetse ingested a bloodmeal six days following a single topical application of NTBC (S11 Fig). To address the opposite situation where blood-feeding precedes topical exposure to NTBC, the drug was topically applied at

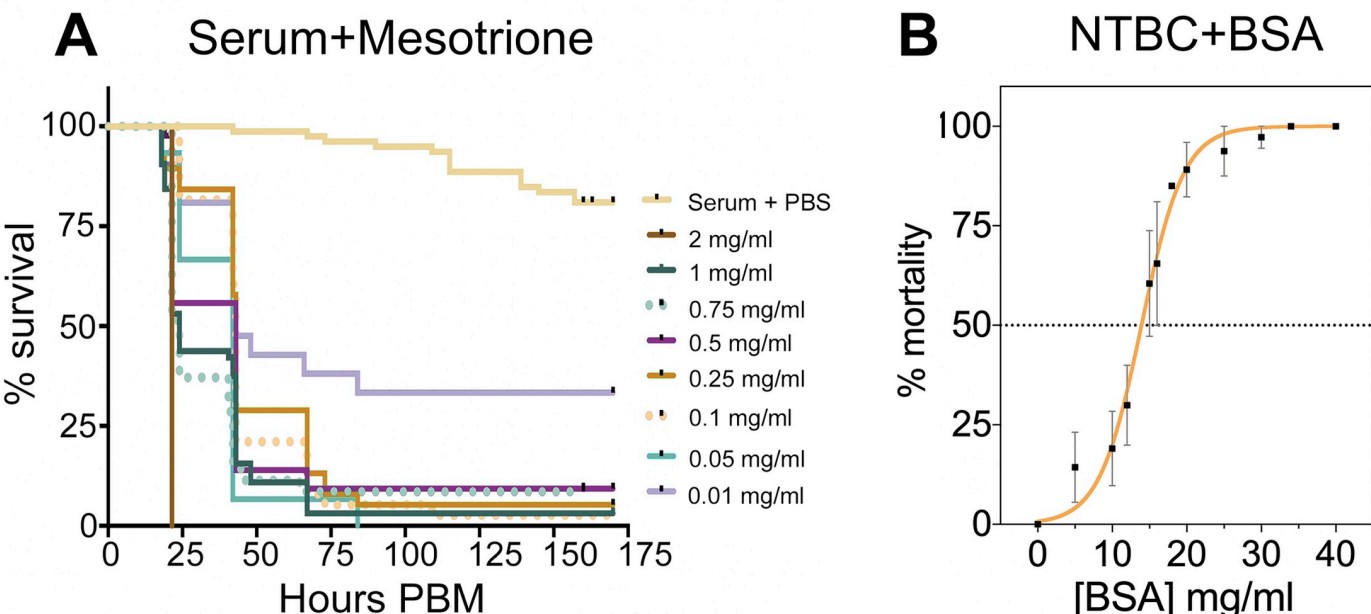

**Fig 4. NTBC-associated toxicity depends on the concentration of protein in the bloodmeal.** (A) Percent survival of flies fed with horse serum supplemented with PBS (control) or different mesotrione concentrations. Three independent experiments were performed: $n$ = 10–25 tsetse per treatment (355 insects in total). (B) Tsetse mortality (percentage) was calculated after feeding flies with different concentrations of BSA as a protein source alongside a lethal concentration (0.001 mg/ml) of NTBC. A protein concentration of 12 to 15 mg/ml is required to induce NTBC lethality (dotted horizonal line, $LC_{50}$). Four independent experiments were performed; each BSA concentration tested used $n$ = 10–80 flies (788 insects in total). Data are shown as mean ± SEM. The underlying data for this figure can be found in S1 Data. BSA, bovine serum albumin; NTBC, nitisinone; PBM, post-blood meal.

different times post-bloodmeal. NTBC reduced tsetse survival up to 48 hours after a bloodmeal (S12 Fig), which implies that bloodmeal proteins were digested after two days (as predicted for tsetse) and excess tyrosine had been catabolised. In contrast to NTBC, topical application of mesotrione was not lethal to tsetse, which could be due to reduced penetration through the insect cuticle (S13 Fig).

## NTBC is not metabolised by insect P450 enzymes

Metabolic resistance due to increased rates of insecticide metabolism by P450s can cause resistance liabilities for new compounds. However, NTBC appears to only be moderately metabolised by CYP3A4 in humans [27], with little oxidative metabolism by other liver CYP enzymes [28]. We have incubated NTBC with microsomes extracted from tsetse, *Aedes*, and *Anopheles* and failed to detect evidence of metabolism as measured by substrate depletion/turnover (S2 Table). Furthermore, we find no obvious metabolism by recombinant *Anopheles gambiae* CYP6P3, a P450 with broad substrate specificity similar to CYP3A4, and associated with cross-resistance to pyrethroids and other insecticides [29]. Overall, this suggests that NTBC may be a weak substrate for P450 metabolism in these insects. This does not preclude the evolution of metabolic resistance, but suggests it may be less likely to emerge rapidly.

## NTBC is highly stable under environmental conditions

Considering uses for NTBC in the field, we examined several stressors that could degrade drug activity and reduce its toxicity. An NTBC solution was subjected to 10 freeze–thaw cycles, prolonged ambient temperature storage (5 weeks), or different exposures to light (S3 Table).

These NTBC test samples (final concentration in blood 3 μM = 1 μg/ml) were screened for activity by adding them into a tsetse bloodmeal as a type of activity bioassay. In all cases, the fly mortality resulting from the stressed samples matched that of freshly made NTBC, thus indicating NTBC does not quickly degrade and is stable under simple, room temperate storage conditions. In agreement with our results, Barchanska and colleagues [30] recently demonstrated that NTBC shows considerable stability under different experimental conditions such as pH of solution, temperature, time of incubation, and ultraviolet radiation.

## NTBC is not toxic to the insect pollinator, *Bombus terrestris*

Bees are the world's most important pollinators of food crops [31], and several reports have shown that bee populations are directly and indirectly affected by insecticides [32]. To determine the environmental impact of NTBC on off-target species (non-bloodfeeders), we investigated the mortality of phytophagous pollinators when exposed to NTBC. Colony-reared *Bombus terrestris* (the buff-tailed bumblebee) were fed ad libitum with NTBC-supplemented sugar as a sole hydration source; bee mortality was assessed over 10 days. Mortality rates did not differ between NTBC-treated and control bees during this sustained exposure, despite feeding NTBC at doses as high as 50 μg/ml (152 μM) and providing pollen as a protein source (Fig 1G).

## A mathematical model supports the use of NTBC for tsetse control

To illustrate the impact that NTBC may have in controlling African trypanosomiasis, we include human and livestock treatment in an epidemiological model and simulate the resulting reduction in transmission (details in S1 Methods). Fig 5 shows what NTBC treatment regime is required to interrupt transmission for systems with diverse vector biting ecologies. The temporal dynamics associated with control are shown in S14 Fig. Fig 5 is generated by averaging the disease control achieved over the time between doses. When both livestock and humans receive treatment (at 80% coverage), monthly dosing can control disease spread if at least 20% of bites are on humans. If fewer bites are on humans, this only becomes a feasible stand-alone control strategy when employed at higher frequencies (e.g., every 10 days). When wildlife (preferred host for some tsetse species) comprises a substantial blood source, treatment regimens become less tenable. At the current plasma efficacy half-life, NTBC will be most useful as part of an integrated disease control strategy. Simulations were also repeated for a transmission system in which animal reservoirs play a negligible role such as in many regions endemic for trypanosomiasis (*T. b. gambiense)* where humans are the primary reservoir. Here, absence of the animal reservoir facilitated disease control when humans comprised at least 20% of tsetse fly bloodmeals and NTBC dosing was monthly (S15 Fig).

## Discussion

When blood-sucking insects digest a bloodmeal, large amounts of tyrosine are produced, which would be potentially toxic if it were not for the first two enzymes in the tyrosine degradation pathway, TAT and HPPD [11]. Here we provide evidence that tyrosine detoxification is essential for tsetse survival post-bloodmeal and have further investigated the possible mechanisms underlying this lethal phenotype. Furthermore, two commercially available HPPD inhibitors were evaluated as novel tsetse control interventions: mesotrione and NTBC. We concluded that NTBC is the best candidate for future field trials due to its low-dose efficacy and environmental sustainability. This drug presents unique features when compared with current insecticides as it specifically targets blood-feeding insects, has low toxicity to mammals, and can be used effectively against tsetse via two delivery routes. NTBC kills tsetse upon

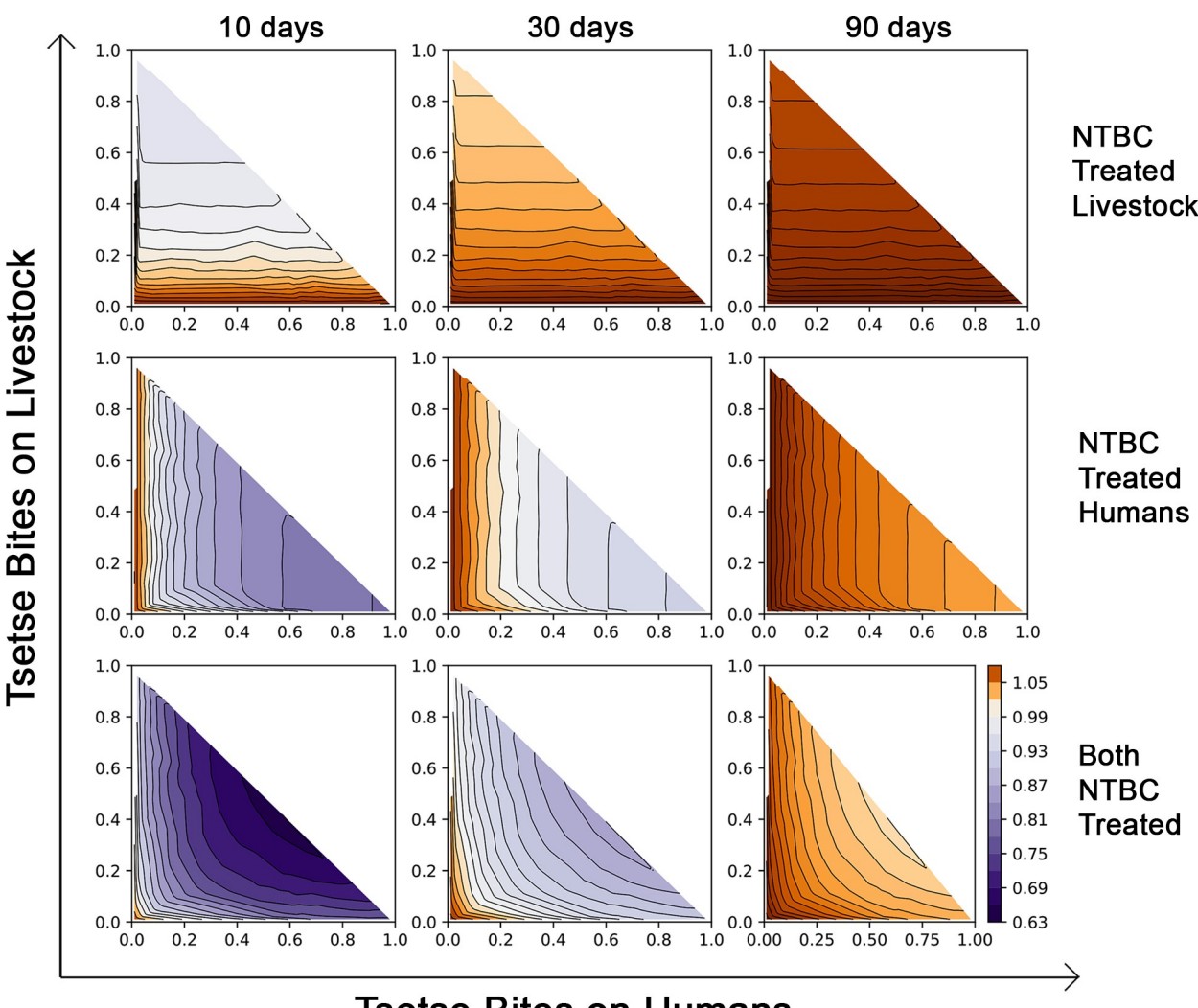

**Fig 5. Impact of NTBC application on the effective reproduction number of *T. brucei*, ($R_e > 1$ in brown and $R_e < 1$ in purple).** Top row: NTBC applied to livestock only; Middle row: Only humans are treated with NTBC; Bottom row: Simultaneous treatment of both livestock and humans with NTBC. Treatments are modelled with NTBC application every 10 days (left column), every 30 days (middle column), or every 90 days (right column). The axes denote the proportional split of bloodmeals taken by local tsetse populations when humans, livestock, and wildlife are potential hosts (where proportion of bites on wildlife are 1 - (bites on humans + bites on livestock)). NTBC, nitisinone.

cuticular application, and it can be safely administered to a mammalian host as an endectocide to kill *Glossina* upon bloodfeeding (and other hematophagous arthropods feeding on the treated mammal [11]). In agreement with our results on tsetse, a recent preprint article from the Oliveira's group has shown evidence that NTBC is more potent than mesotrione and isoxaflutole against *Aedes aegypti* mosquitoes [33]. Moreover, NTBC was also found to be lethal against *Anopheles* and *Culex* as well as *Ae. aegypti* strains resistant to neurotoxic insecticides [33], indicating that cross-resistance between these compounds and NTBC is unlikely.

Our mathematical model indicates that when both livestock and humans are treated with NTBC as an endectocide (at 80% coverage), drug administration every 30 days would control the spread of African trypanosomiasis so long as >20% of the tsetse were feeding on humans. In the absence of an animal reservoir, as it is the case for *T. b. gambiense*, facilitated control could also be achieved with monthly dosing. Transmission of gambiense HAT has been

effectively reduced over the last decade with the deployment of tiny targets in many disease endemic areas [34]. However, disease reactivation remains a concern due to the potentially high number of asymptomatic individuals carrying tsetse-transmissible parasites in the skin [35–38]. Nevertheless, before mass treatment of humans with NTBC is considered, caution should be exercised; it should only be used in regions where trypanosomiasis transmission continues despite best efforts in case management.

Mass drug administration (MDA) for vector control has shown great promise in reducing disease burden [22]. However, MDA raises several issues as its possible long-term benefits are uncertain. The efficacy of MDA approach relies on the level of community participation and also depends on balancing individual and public health interest and potentially limiting individual autonomy by making MDA compulsory [39]. Ivermectin is currently the main drug used as an endectocide; it reduces the survival of most hematophagous vectors when ingested, including tsetse (see below). However, ivermectin's half-life in human blood ranges between 12 to 36 hours and requires the use of multiple doses [21]. Other compounds, such as the isoxazoline veterinary drugs, have been also evaluated for their possible use as endectocides. However, these compounds are only approved for veterinary use and their activity and toxicology profiles in humans have not yet been evaluated [23]. In contrast, the toxicology, pharmacokinetics, and metabolism information is well known for NTBC, and additionally, gained safety approval to be used to treat children since the early 1990s [17,28].

Ivermectin has been evaluated *in vivo* as a potential tsetse control tool. *Glossina palpalis gambiensis* fed on cattle treated with a single ivermectin dose showed decreased tsetse survival ranging from 21% to 84% for the therapeutic dose (0.2 mg/kg), and from 78% to 94% for a dose 10 times greater (2 mg/kg) [40]. However, it is important to highlight that another tsetse species, *Glossina palpalis palpalis*, remained unaffected by an ivermectin dose of 2 mg/kg (4% mortality after 25 days) [41], and no mortality was observed when flies fed on ivermectin-treated (0.5 to 1.0 mg/kg) guinea pigs and goats [42]. Similarly, no effect on fly mortality was observed when *Glossina tachinoides* fed on ivermectin-treated pigs [43]. These differences may be due to pharmacokinetic variations in the different host species (guinea pigs, goats, cattle, and pigs) rather than *Glossina* spp.-specific susceptibilities. In our study, approximately 90% of the treated *G. pallidipes* died within 26 hours after ingesting a single bloodmeal from NTBC-treated rats that had received 1 mg/kg. The surviving flies were severely compromised, had lost the ability to fly, and failed to recover with time. Compared to published data on ivermectin-induced tsetse mortality, dose-matched NTBC is more potent as it kills the flies faster than ivermectin. NTBC is also classified as a safer drug for mammals; the oral $LD_{50}$ for NTBC in rats is $>1,000$ mg/kg [20], while ivermectin is $>100$ times more toxic presenting a $LD_{50} = 10$ mg/kg [44].

As observed in the different dose-response curves, it is not only the concentration of drug ingested or applied topically that is important, but also the quantity of blood ingested and the fly's digestion rate of bloodmeal proteins. This is due to the particular mode of action of HPPD inhibitors. These drugs are not toxic to hematophagous arthropods during starvation or when insects are fed with low-protein content diets, but their toxicity is caused by the accumulation of tyrosine derived from hydrolysis of dietary proteins. This implies that any factor affecting bloodmeal digestion, such as temperature, may also affect HPPD inhibitor toxicity and consequently how long the bloodfed arthropods take to die. Although we discarded flies that did not feed to prevent data dispersion, young, newly emerged flies often take a partial bloodmeal, which likely explains why a small proportion of flies survived even at high drug concentrations. In the field, flies taking partial bloodmeals are also possible (interrupted feeding on host), and, consequently, this may reduce the immediate killing efficacy of any endectocide intervention.

Our data show that protein (but not haem) is sufficient to trigger the NTBC-induced lethal phenotype in tsetse. Together with metabolomic data, these results point to tyrosine accumulation (and precipitation in haemocoel and tissues) as the primary cause of insect death. Feeding flies with sugar meals supplemented with NTBC and BSA was lethal to tsetse and confirmed that tyrosine accumulation from dietary protein digestion is likely the cause of the lethal phenotype. It is worth noting that the quantity of protein to be lethal upon HPPD inhibition may depend on the tyrosine content of each protein; BSA tyrosine content is 3.46% (21/607 residues). Considering that NTBC is very stable under different environmental conditions, these results expand the possibility of also using NTBC-protein mixes in attractive targeted sugar baits (ATSBs) to control different vector-borne diseases.

When assessing the feasibility of topically using NTBC to control tsetse flies, a high mortality was still observed when flies were fed up to 144 hours after NTBC topical application (S11 Fig). The longevity of this drug activity is striking and very encouraging for field-based vector control interventions. In lab-reared colony flies, most tsetse die of dehydration/starvation after such an extensive 144-hour starvation period and, of those remaining alive, many are too weak to feed when offered a bloodmeal. In wild tsetse, where fly activity and nutritional demands are greatly increased, flies are less likely to survive 144 hours of starvation so the sustained lethality of NBTC is even more encouraging. When comparing tsetse lethality to deltamethrin, NTBC is far less potent when topically applied to tsetse. This is not surprising as deltamethrin is a neurotoxin, while NTBC acts by a slower mechanism linked to amino acid catabolism. The disadvantages using deltamethrin for vector control are significant; potency comes at a price. Deltamethrin has a higher toxicity profile to mammals and because it nonselectively kills, it draws a heavy environmental penalty. All insects including pollinators and other non target species are indiscriminately killed. Additionally, deltamethrin causes severe toxicity in aquatic ecosystems [45]. Despite this, pyrethroid insecticides remain the most efficacious vector control strategy for many hematophagous arthropod populations, although increasing reports of insecticide resistance compromise their efficacy [7,46–48]. In contrast, the NTBC toxicity profile is low (high doses are tolerated in humans and other mammals [20]), and it selectively kills blood-feeding arthropods, thus making it environment-friendly and more socially and ethically acceptable to incorporate into vector control strategies. Since NTBC targets a pathway separate to the neurotoxic insecticides and antiparasitic drugs, such as pyrethroids and ivermectin, respectively, its use could complement current vector control tools and reduce the emergence of insecticide-resistant populations.

As previously was reported in humans [28], we found that NTBC is not metabolised by insect P450 enzymes. Importantly, reversible competitive inhibitors of HPPD bind to this enzyme in the same active site that binds its substrate, HPPA [49–52]. Mutations that would affect the binding of NTBC to HPPD may also alter the affinity for HPPA. Due to the importance of HPPD in the physiology of hematophagous arthropods, mutations that reduce its affinity for HPPA would probably be lethal. However, we cannot rule out the existence of some genetic variation or phenotypic plasticity that could confer tolerance to hematophagous vectors towards HPPD inhibitors. Common agriculture-based HPPD inhibitors like mesotrione have been used as herbicides for 20 years, and the emergence of resistant weeds has been reported [53,54]. Interestingly, sequencing the HPPD gene from sensitive and resistant plants showed no target-site mutations that could be associated with resistance to mesotrione [53,54]. Furthermore, no gene duplication or overexpression of HPPD, before or after herbicide treatment, was detected. In contrast, higher levels of mesotrione metabolism via 4-hydroxylation of the dione ring [53] and an enhanced rate of herbicide metabolism [54] were observed in resistant plants. Moreover, a new gene called *HIS1* (HPPD Inhibitor Sensitive 1) was found to confer resistance to several triketone herbicides (including mesotrione) in rice

plants [55], but our Vectorbase search (www.vectorbase.org) did not find orthologous genes in the genomes of several insects, thus indicating the absence of this gene in invertebrates. In summary, two important mechanisms that commonly confer resistance to insecticides (P450s-mediated detoxification and target-site mutations) are unlikely to generate resistance towards HPPD inhibitors in insects. However, since insects are capable of evolving multiple resistance mechanisms, they would could eventually develop resistance to NTBC if used as monotherapy.

We conclude that both NTBC and mesotrione induce rapid death in bloodfed tsetse, although the potency and the duration of their effects are very different (as they are in humans [17]). For these reasons, we propose using NTBC (but not mesotrione) to develop new and complementary vector control tools that target tsetse and other blood-feeding arthropod populations. The importance of using integrated approaches including novel vector controls to achieving elimination was one of the general conclusions of the third WHO meeting of stakeholders on the elimination of gambiense HAT in Geneva, 2018 [56]. The versatility of NTBC is also noteworthy; this FDA-approved drug could be used as a stand-alone technology to control tsetse and other vector populations as an endectocide, or it could complement (or provide an alternative to) the use of standard insecticides. Thus, NTBC is a safer and more environment-friendly alternative to neurotoxic insecticide-based vector control as it is not toxic to mammals and selectively kills blood-feeding arthropods.

## Materials and methods

### Ethics statement

Male (4 to 6 weeks old, 180 to 250 g in weight) Wistar rats sourced from the Animal Rearing Unit and Containment Unit (ARCU) at International Centre of Insect Physiology and Ecology (*icipe*) were used. The animals were housed in standard plastic rodent cages (Thoren Caging Systems, Hazleton, United States) with wood shavings as bedding material. The rodents were maintained on commercial food pellets (Unga Kenya, Nairobi, Kenya), and water was provided *ad libitum*. Animal use and all accompanying procedures and protocols were in accordance with *The Guide for the Care and Use of Laboratory Animals* (Institute for Laboratory Animal Research, 2011). These procedures and protocols were reviewed and approved by Institutional Animal Care and Use Committee (IACUC) of *icipe* (Ref. No. IcipeACUC2018-003).

### Tsetse

The *G. m. morsitans* Westwood colony (originally from Zimbabwe) was housed in an insectary at the Liverpool School of Tropical Medicine. Flies were kept at 26˚C and 73% ± 5% relative humidity and a 12-hour light:dark cycle. The colony was maintained on defibrinated horse blood (TCS Biosciences, Buckingham, United Kingdom) and fed every 2 to 3 days using artificial silicon feeding membranes. Male and female experimental flies of different ages were separately caged and fed. The *G. pallidipes* (originally from Nguruman, Kenya) at the Insectary Unit at *icipe* were maintained at 26˚C and 75% ± 4% relative humidity with 12-hour light:dark cycle. The flies were fed on defibrinated bovine blood using an *in vitro* silicon membrane system. Only teneral *G. pallidipes* were fed on NTBC-treated rats.

### Bumblebee rearing

Hives of *Bombus terrestris audax* ordered from Agralan (Wiltshire, United Kingdom) were kept at 27˚C under constant red light and fed *ad libitum* with pollen and 50% sugar water (Ambrosia syrup, EH Thorne). Only worker bees were used in this study. Five bees from five

different colonies were placed into an acrylic box in five biological replicates, totalling 25 bum-blebees per each experimental group. NTBC was diluted in sugar solution and placed into the box 24 hours after the bees acclimatised to laboratory conditions. Bees were allowed to continually drink NTBC-supplemented water *ad libitum* for 10 days. Each group received PBS or a dose of NTBC at concentrations shown to be lethal to tsetse (0.05 mg/ml, 0.005 mg/ml, or 0.0005 mg/ml). Solutions were changed every five days. Worker bees were also provided with pollen (protein source) according to their rate of consumption. Bee mortality was scored daily for 10 days.

## Phylogenetic analysis

Dendrograms showing the phylogenetic relationships of TAT and HPPD proteins among insects were created according to the maximum likelihood method. Confidence values for each branch were determined through bootstrapping at 100. Analysis was performed on the web-based interface on Phylogeny.fr [57,58], using T-Coffee for alignments under the BLOSUM62 scoring matrix and optimised with Gblocks. Dendrograms were constructed with PhyML 3.0 [59]. The TAT and HPPD protein sequences used in the phylogenetic analyses were taken from either Vectorbase or NCBI using the most recent gene sets available.

## Synthesis of double-stranded RNA (dsRNA)

Specific primers for *G. m. morsitans* TAT and HPPD genes were designed using primer-blast software (NCBI, https://www.ncbi.nlm.nih.gov/tools/primer-blast) (Table 1). These primers contained the T7 polymerase binding sequences required for dsRNA synthesis at 5′ end. The green fluorescence protein (GFP) gene amplified from the peGFP-N1 plasmid (NCBI ID: U55762.1) was used as a control dsRNA to assess *in vivo* off-target effects. PCR products were sequenced to confirm identity. MEGAscript High Yield T7 Transcription kit (Ambion, Austin, United States) was used according to the manufacturer's instructions to synthesise dsRNA. After synthesis, dsRNAs were precipitated by adding an equal volume of ice-cold isopropanol, and the resulting pellets were washed with ethanol, air dried, and then the pellet resuspended in ultrapure water. The dsRNAs concentrations were determined spectrophotometrically at 260 nm on a NanoVue Plus Spectrophotometer (GE Healthcare, Buckinghamshire, United Kingdom) and visualised in an agarose gel (1.5% w/v) to determine dsRNA size, integrity, and purity. An ISS110 speedvac concentrator (Thermo Scientific, Waltham, United States) was used to dry the dsRNAs and samples were adjusted to a final concentration of 5 µg/µl in sterile nuclease-free water (NFW).

## Gene silencing

The protocols described by Walshe and colleagues [60] were followed for RNAi knockdown in tsetse. Briefly, male *G. m. morsitans* were injected in the thorax with 10 µg of each target gene dsRNA (2 µl of 5 µg/µl stock). Tsetse controls were injected with 10 µg of GFP dsRNA. Injection needles were handmade using pulled micro-haematocrit glass capillary tubes (2.00 mm

**Table 1. Sequences of the primers used to amplify target genes for RNAi experiments.** T7 promoter sequences that were necessary for transcription are underlined.

| Gene | NCBI/Vector Base ID | Forward primer | Reverse primer |
|------|--------------------|----------------|----------------|
| *Green fluorescence protein* | U55762.1 | TAATACGACTCACTATAGGGACGTAAACGGCCACAAGTTC | TAATACGACTCACTATAGGGCTTGTACAGCTCGTCCATGCC |
| *Tyrosine aminotransferase* | GMOY012088 | TAATACGACTCACTATAGGGAGACGAAGTGACTGCCGGTCTACG | TAATACGACTCACTATAGGGAGATTCACGAGGCACTGTTAGCAC |
| *Hydroxyphenylpyruvate dioxygenase* | GMOY012145 | TAATACGACTCACTATAGGGAGAATCGCAGCCAATATCGTGGTG | TAATACGACTCACTATAGGGAGACTTTAATTTTGGTGCGGCTGTGC |

outside diameter) (Globe Scientific, Mahwah, United States) mounted inside 200 μl yellow micropipette tips and sealed with Araldite epoxy glue. Flies were injected 24 hours after receiving a bloodmeal to increase survival rates. Flies were chilled for 10 minutes on ice to immobilise them, and 2 μl of dsRNA was injected into the dorsolateral surface of the thorax. Injected flies were allowed to rest for 24 hours before the next bloodmeal. Following injections, flies were fed every 2 to 3 days on sterile, defibrinated horse blood. Three days after dsRNA injection, the digestive systems were excised to determine the efficacy of gene-silencing as evaluated by quantitative PCR (qPCR).

## RNA isolation and cDNA synthesis

*G. m. morsitans* flies were immobilised on ice, and the digestive system was dissected into ice-cold PBS. The total RNA was extracted using TRIzol reagent (Invitrogen, Carlsbad, United States), according to the manufacturer's instructions. Following treatment of the extracted mRNA with Invitrogen Ambion TURBO DNase (Thermo Fisher Scientific, Loughborough, UK), first-strand cDNA synthesis was performed using 1 μg total RNA with "Superscript III First-strand Synthesis System for RT-PCR Kit" (Thermo Fisher Scientific, Loughborough, UK) and poly-T primer, according to manufacturer instructions. The cDNAs were stored at −80°C until use.

## Quantitative polymerase chain reaction (qPCR)

Gene-specific qPCR primers were designed to amplify a different region from that amplified by the RNAi primers to prevent dsRNA amplification. They were designed to span different exons to easily discern contaminating genomic DNA amplification. Primer efficiency was experimentally tested (Table 2). *Glossina α-tubulin* and *β-tubulin* genes were used as references (housekeeping) genes. Quantitative PCR was performed using a MxPro – Mx3005P thermocycler (Agilent Technologies, Santa Clara, United States) with Brilliant III Ultra-Fast SYBR Green QPCR Master Mix (Agilent Technologies, Santa Clara, CA, USA) under the following conditions: 95°C for 15 minutes, followed by 40 cycles of 95°C for 15 seconds, 60°C for 30 seconds, and 72°C for 30 seconds, and a final extension of 72°C for 10 minutes. Target gene expression levels were assessed as $2e^{-\Delta CT}$ values ($\Delta C_T = C_T$ gene of interest – $C_T$ housekeeping gene) and were used to evaluate the mRNA levels of the genes in the different experimental groups (dsTAT- or dsHPPD-injected flies and dsGFP-injected flies) [61].

## Oral dosing of mesotrione and NTBC

Mesotrione (2-[4-(Methylsulfonyl)-2-nitrobenzoyl]cyclohexane-1,3-dione; PESTANAL analytical standard; Sigma-Aldrich, St. Louis, United States) and NTBC (2-[2-nitro-4-(trifluoromethyl) benzoyl]cyclohexane-1,3-dione; PESTANAL, analytical standard; Merck Life Science UK Limited, Gillingham, UK) were solubilised in sterile PBS (NaCl 0.15 M, Na-phosphate 10 mM (pH 7.0)) to different final concentrations (pH was readjusted to 7.0 with 1 M NaOH). One volume of the drugs, or PBS as control, was then mixed with nine volumes of sterile defibrinated horse blood, and these bloodmeals were fed to male and female flies. Only flies with visible redness in the abdomen were selected, and unfed flies were discarded. Final mesotrione concentrations used in blood were: 2, 1.5, 1, 0.75, 0.5, 0.25, 0.1, 0.05, and 0.01 mg/ml. Final NTBC concentrations in blood were: 0.05, 0.025, 0.01, 0.005, 0.0025, 0.00075, 0.0005, 0.00025, and 0.0001 mg/ml).

## Confocal microscopy

Teneral (<24 hours old post emergence) male tsetse were fed on horse blood (TCS Biosciences, Buckingham, United Kingdom) and three days later were offered a second meal

**Table 2. Sequence of the primers used to amplify target genes by real-time PCR.**

| Gene | Vector Base ID | Forward primer | Reverse primer | % Efficiency |
|---|---|---|---|---|
| *Alpha tubulin* | GMOY004645 | TGTATGTTGTATCGTGGTGATGT | GAATTGGATGGTGCGTTTAGTTT | 100.6 |
| *Beta tubulin* | GMOY000148 | CCATTCCCACGTCTTCAGTT | GACCATGACGTGGATCACAG | 96.6 |
| *Tyrosine amino transferase* | GMOY012088 | CCTAGCAATCCGTGTGGTAGTG | TAACCGCTATGTGCTGCGAAC | 103 |
| *Hydroxyphenyl pyruvate dioxygenase* | GMOY 012145 | CTAAAGAACGTGGAGCAACTGTG | TCCACAAAAGTGTGAGTCGT | 106.5 |

containing horse serum (to increase tissue visibility by removing red blood cell interference) with or without 500 ng/mL NTBC. Fifteen hours post-feeding, tsetse were anaesthetized on ice, and the midgut tissue (with attached proventriculi) was dissected into ice-cold PBS and immediately fixed in fresh 4% paraformaldehyde for 1 hour at room temperature (RT). Tsetse group survival rates were determined 72 hours post-NTBC administration. Fixed tissues were washed in PBS (and stained with SiR-actin (1,100 dilution, Cytoskeleton Inc.) for 3 hours at RT and posteriorly incubated in 500 ng/mL DAPI for 10 minutes at RT. Tissues were finally suspended in 1% (w/v) low-melting agarose at approximately 40°C mixed with Slowfade Diamond oil (Molecular Probes, Eugene, United States) on a slide. Slides were imaged using a Zeiss LSM-880 confocal laser scanning microscope and tissues were 3D-reconstructed from a series of z-stacks at intervals of 1.3 μm (Zeiss Company, Oberkochen, Germany).

## NTBC treatment of trypanosome-infected flies

Newly emerged (24 to 48 hours post-eclosion), teneral male *G. m. morsitans* were fed an infected bloodmeal containing fly-infective *T. b. brucei* (strain Antat 1.1 90:13) [62] at $10^6$ parasites per ml of blood. Newly emerged colony-reared flies are highly susceptible to trypanosome infections. Nine days after infection, when procyclic trypanosomes should be established in the midgut, the flies were fed with blood supplemented with NTBC as indicated above.

## Oral dosing of mesotrione in serum

Horse red blood cells were removed from defibrinated horse blood by centrifugation at 1,000 × *g* for 5 minutes. The horse serum was collected, supplemented with mesotrione as indicated above and subsequently fed to female and male *G. m. morsitans*.

## Feeding fructose supplemented with BSA and NTBC

Teneral, male *G. m. morsitans* were sorted into 10 cages and starved for 48 hours. BSA fraction V (stock concentration at 200 mg/ml (w/v) in PBS) was serially diluted into sterile 0.1% (w/v) fructose in PBS to create final concentrations of 40, 34, 30, 25, 20, 18, 16, 15, 12, 10, 5 and 0 mg/ml BSA. Control flies were fed with either diluent alone (0.1% fructose) or diluent plus NTBC. All experimental groups were fed a NTBC dose of 0.001 mg/ml (lethal when delivered with blood or serum). Flies were offered their first meal at 72 hours post-emergence, and mortality was tracked daily for one week.

## Administering HPLA to tsetse by feeding or injection

Newly emerged, male *G. m. morsitans* were sorted into 10 cages containing 25 flies/cage. At 48 hours post-emergence, each cage was fed on defibrinated horse blood supplemented with distilled water (control) or a serial dilution of DL-p-hydroxyphenyllactic acid (HPLA: Aldrich-Sigma, St. Louis, United States). All dilutions were made with the same ratio: 25 μl of additive to 1.975 ml of blood. The concentrations of HPLA tested were 0.05, 0.025, 0.01, 0.005, 0.0025,

0.0075, 0.0005, 0.00025, and 0.0001 mg/ml. All dilutions were kept on ice until feeding to avoid chemical degradation. Fresh HPLA stock concentration was 2 mg/ml in sterile water. After 3 days, flies were offered a normal bloodmeal to test if HPLA impaired a second feed. To administer HPLA by injection, 1-week old male flies that had taken 3 previous bloodmeals (to ensure good health and adequate hydration) were separated into 10 cages. Each group received 2 µl of the selected HPLA concentration via thoracic injection. Stock HPLA was 2 mg/ml, and the concentrations injected were 1.0, 0.5, 0.1, 0.05, 0.01, 0.005, 0.001, 0.0005, and 0.0001 mg/ml with sterile water injected as a control. Flies were allowed to recover for 24 hours and then offered a normal bloodmeal. Fly mortality was monitored throughout in both experiments for a week after HPLA administration, and no group exceeded 10% mortality.

## *In vivo* oral administration of NTBC to rats

Rats received an oral dose of NTBC solubilised in sterile PBS by gavage. The doses administered were 0.1, 0.2, 0.5, 1.0, and 2.0 mg/kg, with controls receiving an equal volume of PBS. After 90 minutes, the rats were anaesthetized with an intraperitoneal injection of 80 µl of 3% ketamine and Xilazin 0.33% diluted in sterile PBS. Ketamine and Xilazin do not affect tsetse mortality. Groups of *G. pallidipes* flies were fed on either PBS- or NTBC-treated anaesthetized rats.

## Topical application of mesotrione and NTBC to tsetse cuticle

Mesotrione and NTBC were solubilised in 100% acetone at the following concentrations (Mesotrione: 10, 5, 2.5, 1, 0.5, and 0.1 mg/ml. NTBC: 1, 0.5, 0.25, 0.125, 0.0625, 0.03175, 0.0156, 0.0078, and 0.0039 mg/ml). One microliter of solution was topically applied with a P2 micropipette to the ventral surface of the abdomen of a cold-anaesthetized *male G. m. morsitans*, either immediately before ingesting a blood meal, at different times before or after a bloodmeal. The control flies received 1 µl of 100% acetone. Only fully engorged flies were used.

## Metabolomic analysis

Male *G. m. morsitans* were dissected at different times (0, 5, or 10 hours) after ingesting horse blood supplemented with NTBC 0.01 mg/ml or PBS. The flies were chilled for 10 minutes at 4˚C and kept on ice. The legs of 20 flies were severed and collected into 1.5 ml plastic tubes containing 300 µl of 0.9% NaCl and 13.34 µM phenylthiourea (internal standard that prevents melanization) in ultrapure water. The samples were vortexed for 1 minute, centrifuged for 10 minutes at 14,000 rpm to collect the supernatant and then syringe-filtered (Millipore 0.22 µm syringe filter) to remove cells, such as haemocytes, present in the haemolymph. Filtered samples (150 µl total volume) were transferred to a 1.5 ml tube containing 50 µl of chloroform, 75 µl of methanol, and 75 µl of water containing B-methylamino-L-Alanine (53.4 µM) as an internal standard for normalisation. After centrifugation, the polar and apolar phases were collected and dried in a speed vac concentrator (ISS100, Thermo Scientific, Dreieich, Germany), and samples were stored at −80 until analysis. Five samples in total were collected from 2 independent experiments for each group and time point.

## LC-MS and LC-MS/MS analysis

Polar extracts were reconstituted in Methanol:Water (1:1, V/V) containing 5 µM 13C5, 15N-Valine as an internal standard. Liquid chromatography-tandem mass spectrometry (LC-MS/MS) was performed as described previously [63]. LC analysis was performed using a

Dionex UltiMate LC system (Thermo Fisher Scientific, Loughborough, UK) with a ZIC-pHI-LIC column (150 mm × 4.6 mm, 5 μm particle, Merck Sequant, Umea, Sweden). A 15-minute elution gradient of 80% Solvent A (20 mM ammonium carbonate in Optima HPLC grade water, Sigma Aldrich) to 20% Solvent B (acetonitrile Optima HPLC grade, Sigma Aldrich) was used, followed by a 5-minute wash of 95:5 Solvent A to Solvent B and 5-minute re-equilibration. Other parameters were as follows: flow rate 300 μl/min; column temperature 25˚C; injection volume 10 μl; and autosampler temperature 4˚C.

All metabolites were detected across a mass range of 70 to 1050 *m/z* using a Q Exactive Orbitrap instrument (Thermo Fisher Scientific, Loughborough, UK) with heated electrospray ionisation and polarity switching mode at a resolution of 70,000 (at 200 *m/z*). MS parameters were as follows: spray voltage 3.5 kV for positive mode and 3.2 kV for negative mode; probe temperature 320˚C; sheath gas 30 arbitrary units; and auxiliary gas 5 arbitrary units. Pooled biological quality control (PBQC) samples were analysed throughout the run to provide a measurement of the stability and performance of the system. Parallel reaction monitoring (PRM) was used at a resolution of 17,500 to confirm the identification of metabolites; collision energies were set individually in high-energy collisional dissociation (HCD) mode. Data were acquired using Xcalibur 3.0.63 (Thermo Fisher Scientific, Loughborough, UK), and Progenesis (Nonlinear Dynamics, Newcastle, United Kingdom) was used for data alignment and peak detection. Data were normalised against internal standards. Metabolites were considered significantly altered using 2-way ANOVA ($p < 0.05$) and Bonferroni post hoc test. Annotations were assigned to accurate masses with a maximum error of 5 ppm using Metlin [64], LipidMaps [65], Kegg [66], and HMDB [67], which were searched simultaneously using the CEU Mass Mediator engine (http://ceumass.eps.uspceu.es/mediator/)). Features that did not present any hint in the database were filtered out. Metabolomics data (five replicates/time point and condition) was deposited into the Metabolights repository (https://www.ebi.ac.uk/metabolights/) under accession number MTBLS2166.

## Metabolomic pathway analysis and heat map

The mean relative abundances of all identifiable metabolites for each time point for both treated and untreated flies were divided by the mean abundance of the metabolite at the corresponding time point from untreated flies. Results were expressed in fold change values for each individual metabolite, with the untreated group serving as the baseline. Thus, metabolites with abundance fold changes greater than 1 were said to be up-regulated in flies treated with NTBC as compared to control, while metabolites with abundance fold change less than 1 were said to be down-regulated.

The abundance fold-change values were then plotted onto a heatmap using gplot's heatmap.2 program (version 3.0.1.1, http://cran.r-project.org). The heatmap colour scale was limited from 0 to 4, with additional non-cosmetic options as follows: distfun = function(x) dist(x, "manhattan"), hclustfun = function(x) hclust(x, "centroid"), Colv = FALSE.

Subsequent to this, metabolic pathways were assembled using Pathway Collages from BioCyc [68,69], focusing on pathways relevant to tyrosine degradation, carbon metabolism, and amino acid biosynthesis. Fold changes from all identified metabolites were overlaid onto this "combined" metabolic pathway, and statistically significant metabolites (as determined by the 2-way ANOVA test as described previously) were highlighted. Metabolites that were identified from LC-MS/MS but were not identifiable within the "combined" metabolic pathway were not mapped. Pathways that did not have any LC-MS/MS-identified metabolites mapped to it were removed from the "combined" pathway.

### *In vitro* comparative metabolism of NTBC against insect microsomal preparations

**Microsome preparation.**    Microsomes were prepared from young female mosquitoes (48 to 72 hours old) and from the mosquitoes (*An. gambiae (*Kisumu) and *Ae. aegypti* (New Orleans) as described by Kasai and colleagues [70] and Inceoglu and colleagues [71] with some modifications. Briefly, preparation of the adult mosquito microsomes (approximately 200 individuals per colony) were done by removing the heads to avoid enzyme inhibition by xanthommatin eye pigments [72]. Mosquitoes were snap-frozen in liquid nitrogen with a sieve approximately 2 mm mesh size. Small steel spheres were gently mixed with the mosquitoes to fractionate their bodies and to separate the heads, legs, and wings parts from the joint abdominal-thoracic components. The thorax and the abdomen were then washed with 3 ml of pre-chilled 0.1 M KBP (pH 7.4) homogenisation buffer (HB) containing 1× protease inhibitor (Roche Complete ULTRA, Basel, Switzerland). The complex was homogenised in 20 ml of HB using 40 ml glass Dounce homogeniser with a loose B pestle (Wheaton Science, Millville, United States) for 20 strokes. The separation of the homogenate into cytosolic and microsomal extracts was achieved by centrifugation steps at 4˚C. The initial centrifugation was performed at $10,000 \times g$ for 15 minutes to remove insoluble debris. A second centrifugation at $200,000 \times g$ for 1 hour was performed to pellet the microsomes. The microsomal pellet was resuspended in 2 ml of ice-cold suspension buffer (HB containing 20% glycerol). Protein concentration was measured in triplicate by the Bradford method [73] with Bio-Rad reagents using BSA as a protein standard.

## Tsetse microsomes

Virgin female *G. m. morsitans* ($n = 174$) were maintained on three bloodmeals over the week and then starved for six days to reduce bloodmeal contaminants during tissue homogenisation. The intact thorax and abdomen were removed by a pair of forceps, snap frozen in liquid nitrogen, and homogenised as described with mosquito microsomes. Because of the low P450 content in insecticide-susceptible tsetse flies and mosquitoes, it was not possible to determine P450 content by the traditional carbon monoxide-reduced spectral assay [74], thus protein concentrations were adjusted to 4 mg/ml using 0.1 M Potassium Phosphate buffer (pH 7.4) to normalise for protein content. Recombinant *An. gambiae* CYP6P3 enzyme was used as a positive control for P450s microsomal activities. All microsome preparations, in comparison to CYP6P3, were tested for O-dealkylation activities against the P450's generic substrate "Diethoxyfluorescein" before the setup of the NTCB metabolism assay that identifies P450s enzyme activities in microsomal extracts. Briefly, for activity assays, 200 μl of reaction media were set up in triplicate containing 0.5 mM NADPH, 5 μM diethoxyfluorescein, 4 mg/ml microsomes or with mixture of 0.1 μM CYP6P3, 0.8 μM cytochrome *b5* and incubated at 30˚C for 15 minutes. The enzyme activity was recorded versus negative control (no NADPH) as $147.93 \pm 32$, $408.64 \pm 45.96$, $51.5 \pm 2.63$, and $372 \pm 9.8$ relative fluorescence units (RFU) for tsetse, *Ae. aegypti*, *An. gambiae*, and CYP6P3, respectively. This showed the suitability of microsomes, CYP6P3, and NADPH for running the NTBC comparative metabolism.

**NTBC metabolism by P450 enzymes.**    NTBC oxidation by P450 enzymes were evaluated. The 200 μl reaction media containing 20 μM of each drug, 0.5 mM NADPH, and the enzyme source (4 mg/ml microsomal extract or CYP6P3 0.1 μM ± 0.8 μM cytochrome b5) was incubated for 1 hour. Reactions were initiated by adding NADPH and incubating at 30˚C with shaking (1,200 rpm). Reactions were stopped with 200 μl acetonitrile and analysed for drug peak depletion using High Performance Liquid Chromatography (HPLC) according to Muller and colleagues [29] with some modifications. The chromatographic separation was achieved

using a Thermofisher C18 column using 0.1% phosphoric acid in water and acetonitrile (45:55, v/v). The cutoff value of 20% substrate depletion used to distinguish true substrate turnover from baseline variability.

### NTBC stability under environmental stressors

NTBC was diluted in 10% (w/v) sucrose to a final concentration of 0.01 mg/ml and stored at either 4°C or 25°C in dark or transparent 1.5 ml polypropylene tubes. One volume of NTBC was then mixed with 9 volumes of sterile defibrinated horse blood (final NTBC concentration: 0.001 mg/ml), and these bloodmeals were used to weekly screen for tsetse lethality throughout a 5-week period.

Additionally, NTBC was diluted in 10% (w/v) sucrose to a final concentration of 0.01 mg/ml and subjected to 10 freeze–thaw cycles. Following temperature stress, one NTBC volume was mixed with nine volumes of sterile defibrinated horse blood (final concentration: 0.001 mg/ml), and these bloodmeals were fed to male tsetse. Mortality was monitored over 24 hours.

### Mathematical modelling

A discrete time (one-day time step) compartmental model was constructed to describe the key processes underlying the transmission of African trypanosomiasis. These parasites have a broad reservoir of host animals, so these were categorised according to whether they were livestock or wildlife. The transmission between three host types in total (humans, livestock, and wildlife), and the tsetse fly vector was simulated for *Trypanosoma brucei rhodesiense* (S1 Methods). For *T. b. gambiense*, animal reservoirs likely play a reduced role, so simulations focused on transmission between tsetse flies and humans only. In both cases, the pre-intervention $R_0$ was assumed to have a value of 1.1 [75,76]. The impact of vector biting behaviour was examined by apportioning the split of bites on humans, livestock, and wildlife randomly across the full possible range (500 iterations) and, through simulation, determining the frequency of NTBC application required to drive the effective reproduction number (*Re*) below unity. See S1 Methods for further details about the mathematical modelling.

### Statistical analysis

Tsetse survival was scored daily post-treatment. Statistical analysis and graphics were performed using Prism 6.0 software (GraphPad Software, San Diego, United States). The data from multiple experiments were combined into a single graph. The log-rank (Kaplan–Meier) test was used to evaluate significant differences in survival between the experimental and control groups. Dose-response curves were completed using a plot of nonlinear fit for log of inhibitor versus normalised response (variable slope). $LD_{50}$ and $LC_{50}$ were calculated using Probit analysis (POLO Plus version 2.0). Numerical source data underlying all figures can be found in S1 Data.

### Supporting information

**S1 Methods. Mathematical modelling for *Trypanosoma brucei* rhodesiense transmission.** Mathematical modelling for *T. brucei gambiense* transmission. (DOCX)

**S1 Table.** Tsetse survival after (A) bloodmeal supplementation with HPLA concentrations or (B) injecting HPLA concentrations into the haemocoel. One independent replicate was performed per experiment; (A) *n* = 25 flies fed/concentration, (B) *n* = 10 flies injected/dose. (DOCX)

**S2 Table. The HPLC depletion assay was used to determine if NTBC is metabolised by** *Glossina* **P450 enzymes (detoxification enzymes).** Summary of *in vitro* NTBC degradation after an hour incubation with insect microsomal preparations and recombinant CYP6P3 in the presence of NADPH.
(DOCX)

**S3 Table. NTBC stability under different environmental conditions (light exposure and storage temperature).** NTBC stock solutions were stored in either opaque (dark) tubes to protect from light exposure or in translucent tubes (light). Tube storage was either at room temperature (RT; 25˚C) or refrigerated (4˚C) over a 5-week time period.
(DOCX)

**S1 Fig. ML phylogenetic trees were created from full-length protein sequences of TAT (A) and HPPD (B) from several insect species.** The species and branches highlighted in orange indicate blood-feeding insects. A partial sequence coding for HPPD was identified for *G. pallidipes*, but it was excluded from the analysis because it was not complete. Scale bars (branch lengths) correspond to the mean number of amino acid substitutions per site on the respective branch. HPPD, 4-hydroxyphenylpyruvate dioxygenase; ML, maximum likelihood; TAT, tyrosine aminotransferase.
(TIF)

**S2 Fig. Efficiency of gene silencing in tsetse midgut tissue three days after intrathoracic injection of dsRNA.** (A) *TAT* (unpaired *t* test with equal SD. $N = 4$–$5$ $p = 0.05$) knockdown. (B) *HPPD* knockdown (unpaired *t* test with equal SD. $N = 5$–$4$ $p = 0.04$). Data are shown as mean ± SEM. dsRNA, double-stranded RNA; HPPD, 4-hydroxyphenylpyruvate dioxygenase; TAT, tyrosine aminotransferase.
(TIF)

**S3 Fig. The survival of tsetse after feeding on blood supplemented with either mesotrione** (A) **or NTBC** (B). Dose-response curves were calculated at 24 hours after the bloodmeal (C). Six independent experiments were performed ($n = 16$–$28$ insects per dose). Total number of tsetse used to generate the dose-response curves were $n = 1{,}018$ for mesotrione and $n = 914$ for NTBC. Data are shown as mean ± SEM. NTBC, nitisinone.
(TIF)

**S4 Fig. No differences in susceptibility to HPPD inhibitors were observed between flies sex. (A)** Mesotrione-fed female tsetse (total insects: 483; LD50: 427.5 μM; 95% CI: 189.5–737.4) and **(B)** Mesotrione-fed male tsetse (total insects: 536; LD50: 304.9 μM; 95% CI: 133.3–507.8). **(C)** Dose-response curves calculated 24 hours after mesotrione feeding. Data are shown as mean ± SEM. **(D)** NTBC-fed female tsetse (total insects: 403; LD50: 2.8 μM; 95% CI: 0.38–8.5). **(E)** NTBC-fed male tsetse (total insects: 511; LD50: 1.7 μM; 95% CI: 0.24–4.32). **(F)** Dose-response curves calculated 24 hours after NTBC feeding. Three independent experiments were performed, each with $n = 26$–$30$ tsetse per dose. Data are shown as mean ± SEM. HPPD, 4-hydroxyphenylpyruvate dioxygenase; NTBC, nitisinone.
(TIF)

**S5 Fig. The percent of** *T. brucei***-infected flies surviving a bloodmeal supplemented with NTBC or PBS.** Two independent experiments were performed, each with $n = 26$–$30$ tsetse per dose (233 insects in total). NTBC, nitisinone; PBS, phosphate-buffered saline.
(TIF)

**S6 Fig. Characterisation of tsetse phenotypes when fed on rats treated with different doses of NTBC.** *G. pallidipes* flies were fed on anaesthetised rats that received an oral dose of sterile PBS (A) or 0.1 mg/kg (B), 0.5 mg/kg (C), or 1 mg/kg NTBC. Survival was recorded for 26 hours post feeding. NTBC, nitisinone; PBS, phosphate-buffered saline.
(TIF)

**S7 Fig. (from S1 Video)** Panels (A) to (C) have control (untreated) tsetse on the left and NTBC-treated (0.001 mg/ml) flies on the right. (A) In both groups, the excretion of water within the first 20 minutes after feeding is due to a process called diuresis. (B) The first fly to be partially paralysed (fly on back, yellow circle) in the NTBC-treated group occurs 8 hours after ingesting the bloodmeal. (C) Evidence of bloodmeal digestion in control flies as evidenced by dark excreta (yellow arrows) that is absent in NTBC-treated flies. (D) Top view of NTBC-treated compartment highlights the characteristic NTBC-treated, blackened abdomens (*). All flies are dead (black eyes) or dying (fully paralysed) by 28 hours post ingestion of NTBC. NTBC, nitisinone; PBS, phosphate-buffered saline.
(TIF)

**S8 Fig.** External and internal tissue destruction of NTBC-treated tsetse: (A) tsetse fed regular blood on left and NTBC-treated blood on the right. Internal abdominal liquefaction was demonstrated by placing tsetse abdomens under a glass slide and applying pressure to squeeze out tissues: (B) control fly and (C) NTBC-treated fly. NTBC, nitisinone.
(TIF)

**S9 Fig. Percent survival of flies fed daily with either 0.1% fructose-PBS or PBS alone supplemented with 34 mg/ml of BSA.** Only a small percentage (approximately 6%) of flies died 24 hours after feeding with either PBS-BSA or fructose-BSA. Moreover, both groups showed a comparable mortality rate until day 6, when they reached approximately 50% of the mortality compared to the group of flies fed only with horse blood. These experiments provide evidence that (1) the 24-hour mortality shown in Fig 4B is due to the addition of NTBC; (2) protein degradation is important for NTBC killing; and (3) tsetse flies do not appear to obtain energy from ingesting 0.1% fructose. Daily feeds were mandatory because the flies are highly susceptible to dehydration as they quickly process the nutritionally poor meal. Fly mortality was daily recorded for a period of six days. Two independent experiments were performed, each with *n* = 18–51 tsetse per treatment (221 insects in total). BSA, bovine serum albumin; NTBC, nitisinone; PBS, phosphate-buffered saline.
(TIF)

**S10 Fig. Dose-response curves calculated 24 hours after topical application of either delta-methrin or NTBC.** Drugs were applied immediately after tsetse had taken a bloodmeal. Three independent experiments were performed: *n* = 10–20 insects per dose. The data for the dose-response curves for deltamethrin (*n* = 339 flies) and NTBC (*n* = 449 flies) are shown as mean ± SEM. NTBC, nitisinone.
(TIF)

**S11 Fig. The percent survival of *G. m. morsitans* after topically applying either acetone (red) or NTBC: 1,000 ng (blue) or 31.25 ng (black).** Once the solution was applied, tsetse were offered a bloodmeal every 24 hours to measure the residual activity of absorbed NTBC. Flies were fed at either 24 hours (**A**), 48 hours (**B**), 72 hours (**C**), 96 hours (**D**), 120 hours (**E**), or 144 hours (**F**) after topical application. Vertical dotted lines indicate the timing of each bloodmeal. Two independent experiments were performed, each with *n* = 10–12 insects (315

insects in total). NTBC, nitisinone.
(TIF)

**S12 Fig. *G. m. morsitans* survival (%) after the topical application of solutions at specific times after an initial bloodmeal (PBM).** Acetone (red) or NTBC (1,000 ng, blue; 21.25 ng, black) was topically applied at (**A**) 24 hours, (**B**) 48 hours, (**D**) 72 hours, and (**D**) 96 hours after a fly had ingested a single bloodmeal. Two independent experiments were conducted, each with *n* = 10–12 insects (306 insects in total). NTBC, nitisinone; PBM, post-blood meal.
(TIF)

**S13 Fig. Tsetse survival after topically applying mesotrione to the fly thorax and then giving them a bloodmeal.** Two independent experiments were performed, each with *n* = 10–30 flies per dose tested (190 insects in total).
(TIF)

**S14 Fig. *T. b. rhodesiense Re* dynamics following NTBC treatment of both livestock and humans every 30 days.** The different lines correspond with randomly selected distributions of bites among humans, livestock, and wildlife. Where wildlife make up the large majority of tsetse bloodmeals, only limited control is achievable (dark brown) but where humans and livestock make up the majority of bloodmeals, good levels of control can be achieved (blue). NTBC, nitisinone.
(TIF)

**S15 Fig. Effect of NTBC application on the effective reproduction number of *T. brucei gambiense* (*Re* > 1 in brown and *Re* < 1 in purple).** Treatments are conducted either every 10 days (left column), every 30 days (middle column), or every 90 days (right column). The axes denote the proportional split of bloodmeals ingested by local tsetse vectors between humans, livestock, and wildlife (where proportion of bites on wildlife are 1 - (bites on humans + bites on livestock)). NTBC, nitisinone.
(TIF)

**S1 Video. Male tsetse flies were fed either normal blood (control) or blood supplemented with a lethal concentration of NTBC (0.001 mg/ml) and immediately transferred into a bespoke Perspex box for filming.** The left compartment contains control flies and the right holds NTBC-treated flies. Initial tsetse knockdown is defined as loss of flight, followed by paralysis and dropping to the floor. The video recorded continually for almost 29 hours and was condensed to 1 minute (100,000 ×); real time is shown in top right corner (hh:mm). Key features, highlighted in yellow, are further described in S7 Fig.
(MP4)

**S1 Data. All experimental raw data used to generate manuscript figures.**
(XLSX)

## Acknowledgments

We thank Daniel Southern, Keith Steen, and Robert Leyland for excellent technical assistance. We thank Robert Prendergast for videography and video editing.

## Author Contributions

**Conceptualization:** Marcos Sterkel, Pedro L. Oliveira, Álvaro Acosta-Serrano.

**Data curation:** Marcos Sterkel, Lee R. Haines, Aitor Casas-Sánchez, Vincent Owino Adung'a, Raquel J. Vionette-Amaral, Shannon Quek, Clair Rose, Mariana Silva dos Santos, Natalia García Escude, Hanafy M. Ismail, Mark I. Paine, Seth M. Barribeau, Simon Wagstaff, James I. MacRae, Daniel Masiga, Pedro L. Oliveira, Álvaro Acosta-Serrano.

**Formal analysis:** Marcos Sterkel, Lee R. Haines, Vincent Owino Adung'a, Shannon Quek, Clair Rose, Mariana Silva dos Santos, Hanafy M. Ismail, Mark I. Paine, Seth M. Barribeau, James I. MacRae, Laith Yakob, Álvaro Acosta-Serrano.

**Funding acquisition:** Álvaro Acosta-Serrano.

**Investigation:** Marcos Sterkel, Lee R. Haines, Aitor Casas-Sánchez, Vincent Owino Adung'a, Raquel J. Vionette-Amaral, Shannon Quek, Clair Rose, Mariana Silva dos Santos, Natalia García Escude, Hanafy M. Ismail, Mark I. Paine, Seth M. Barribeau, James I. MacRae, Daniel Masiga, Laith Yakob.

**Methodology:** Marcos Sterkel, Lee R. Haines, Aitor Casas-Sánchez, Vincent Owino Adung'a, Shannon Quek, Clair Rose, Mariana Silva dos Santos, Natalia García Escude, Seth M. Barribeau, Simon Wagstaff, James I. MacRae, Laith Yakob.

**Project administration:** Álvaro Acosta-Serrano.

**Resources:** Álvaro Acosta-Serrano.

**Supervision:** Vincent Owino Adung'a, Mark I. Paine, Seth M. Barribeau, Simon Wagstaff, Daniel Masiga, Pedro L. Oliveira, Álvaro Acosta-Serrano.

**Validation:** Marcos Sterkel, Lee R. Haines, Aitor Casas-Sánchez, Vincent Owino Adung'a, Raquel J. Vionette-Amaral, Shannon Quek, Clair Rose, Mariana Silva dos Santos, Natalia García Escude, Hanafy M. Ismail, Mark I. Paine, Seth M. Barribeau, Simon Wagstaff, James I. MacRae, Daniel Masiga, Laith Yakob, Pedro L. Oliveira, Álvaro Acosta-Serrano.

**Visualization:** Lee R. Haines, Laith Yakob, Pedro L. Oliveira, Álvaro Acosta-Serrano.

**Writing – original draft:** Marcos Sterkel, Álvaro Acosta-Serrano.

**Writing – review & editing:** Marcos Sterkel, Lee R. Haines, Aitor Casas-Sánchez, Vincent Owino Adung'a, Raquel J. Vionette-Amaral, Shannon Quek, Clair Rose, Mariana Silva dos Santos, Natalia García Escude, Hanafy M. Ismail, Mark I. Paine, Seth M. Barribeau, Simon Wagstaff, James I. MacRae, Daniel Masiga, Laith Yakob, Pedro L. Oliveira, Álvaro Acosta-Serrano.

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
