## [Editor Report · Decision Letter 0]

29 May 2020

Dear Dr Acosta-Serrano, 

Thank you for submitting your manuscript entitled "Repurposing the orphan drug nitisinone to control the transmission of African trypanosomiasis" for consideration as a Research Article by PLOS Biology. I apologize for the delay in getting you this decision. As I’m sure you can understand, our Academic Editors (and reviewers) currently have very limited availability due to COVID-19 related disruptions and an increased review load, and our editorial team is also affected. Additionally, we have been seeing a large increase in submissions during this time. Please do accept our very sincere apologies for the unavoidable delays.

Your manuscript has now been evaluated by the PLOS Biology editorial staff. We had been hoping to have an Academic Editor's input as well but have decided to move ahead regardless. The editorial team is very interested in your manuscript and I am writing to let you know that we would like to send your submission out for external peer review.

Please re-submit your manuscript within two working days, i.e. by May 31 2020 11:59PM.

Kind regards,

Hashi Wijayatilake, PhD,

Managing Editor

PLOS Biology

---

## [Decision Letter · Decision Letter 1]

30 Jun 2020

Dear Alvaro,

Thank you very much for submitting your manuscript "Repurposing the orphan drug nitisinone to control the transmission of African trypanosomiasis" for consideration as a Research Article at PLOS Biology. Your manuscript has been evaluated by the PLOS Biology editors, an Academic Editor with relevant expertise, and by several independent reviewers, whose expertise and comment you will find at the end of this email.

As you will see, reviewers 1, 2 and 4 are overall very supportive of the work and we have decided to overrule referee 3's novelty concerns, as we consider that demonstrating the applicability of this approach goes well beyond your previous proof of principle studies. Nevertheless, the referees have identified several key experiments and analyses that need strengthening and would be important to do, especially the modelling and PTU experiments, and request some additional discussion and caveating of the results, and some reworking of the figures. In addition, reviewers 1 and 3 request that data not shown be included in the manuscript, which is also our journal policy and thus must be addressed.

In light of the reviews (below) we would be happy to invite revision of a study that fully addresses the concerns outlined above. We cannot make any decision about publication until we have seen the revised manuscript and your response to the reviewers' comments, which is also likely to be sent for further evaluation by the reviewers.

We expect to receive your revised manuscript within 4 months; please let us know if you foresee that the revision process may be longer. 

**IMPORTANT - SUBMITTING YOUR REVISION**

*Re-submission Checklist*

*Published Peer Review*

*PLOS Data Policy*

*Blot and Gel Data Policy*

With best wishes,

Nonia

Nonia Pariente, PhD, 

Editor-in-Chief PLOS Biology

PLOS Biology

REVIEWS:

Reviewer's Responses to Questions

PLOS authors have the option to publish the peer review history of their article (what does this mean?). If published, this will include your full peer review and any attached files.

Reviewer #1: Yes: Louis Lambrechts (ecology, evolution and genetics of insect-virus interactions)

Reviewer #2: Yes: Geoffrey M Attardo (tsetse physiology and trypanosomiasis control)

Reviewer #3: No (interventions to prevent malaria transmission; mosquito vector control strategies; malaria pathogenesis)

Reviewer #4: No (modelling of parasite spread and effect of anti-parasitic interventions)

Reviewer #1: This is a well-conducted and exciting study on the potential of the FDA-approved drug nitisinone to control populations of hematophagous vectors. The body of evidence provided in the paper convincingly demonstrates that nitisinone (a tyrosine degradation inhibitor) can kill tsetse flies upon blood feeding by promoting toxic accumulation of tyrosine. The potency of the drug at concentrations that are compatible with mass drug administration in the human population, the versatile application (oral or topical) and the environmental friendliness (harmless to bumble-bees) make nitisinone a very promising candidate for drug-based vector control. The paper is clearly written and I only have minor suggestions to improve it, listed below in decreasing order of importance.

It would be interesting to discuss the likelihood that flies (or other insects) could evolve nitisinone resistance by other means than metabolic detoxification (e.g., target insensitivity). In the dose-response experiments, a small proportion (10-15%) of flies seems to survive even at high drug concentration (e.g., Fig. 1d). This observation is consistent with the existence of some genetic variation and/or phenotypic plasticity in nitisinone tolerance.

Since PLoS Biology has no restrictions on the number of figures, some of the data relegated to the supporting information could be included in the main body of the paper. For instance, Figs. S6, S7 and S8 could be combined and provided as an additional main figure.

It is not expected that the authors carry out additional experiments with mosquitoes, but perhaps the collateral effects of nitisinone on mosquito vector species (if they are considered plausible) could be mentioned in the discussion as an additional benefit to the proposed strategy.

Some guidance in the caption could help the unfamiliar reader to navigate the ternary plots shown in Fig. 4. For example, placing the labels at the corners (not the edges) would make it easier to interpret the simulation results. Frequency (in days) should probably refer to time interval between applications.

Line 108: why are the data not shown ?

Reviewer #2: The manuscript by Sterkel et al. describes the the implementation of a tyrosine catabolism inhibitor (nitisinone - NTBC) as a a putative insecticide targeted at control of blood feeding arthropods with tsetse flies being the focus of this work.

The authors demonstrate the essential nature of tyrosine detoxification in blood meal digestion and metabolism as knockout of the genes for key enzymes in the tyrosine detoxification pathway (TAT and HPPD) causes lethality in flies after blood feeding. This is followed up by treatment with inhibitors of these enzymes (mesotrione and NTBC) and demonstrate lethality after blood feeding when treated with these drugs either orally or topically. In particular, nitisinone, a FDA approved drug with a well tolerated mammalian toxicity profile, is effective a low concentrations equivalent to therapeutic dosages found in humans. Tsetse flies feeding on rats treated with therapeutic levels of nitisinone also become intoxicated and die. The results from this paper are exciting and potentially game changing as a way to enable vector control to reduce levels of both human and animal trypanosomiasis.

The manuscript is well written and easy to read, the materials and methods are comprehensive and well described and the data is presented clearly. I feel the manuscript is well suited to PLoS Biology. However, I think some of the statements in the manuscript regarding the data are overinterpreted and either need to be toned down or additional controls/experimental work need to be included to provide support for these statements. I also think the final figure could use some modifications to make it easier to interpret and should include one alternative condition.

The authors perform a thorough analysis of the pathology resulting from in nitisinone treatment revealing extensive disruption and breakdown of internal tissues/organ systems. They demonstrate via metabolomic analysis that treatment with the inhibitor results in accumulation of toxic levels of tyrosine and hydroxyphenylpyruvic acid as a result of the HPPD inhibition. They go on to examine the potential for excess melanization activity and/or haem accumulation to be the cause of pathology. The authors state that they inhibited phenoloxidase (PO) activity with treatment of phenylthiourea (PTU), however, the experiment lacks a positive control to demonstrate the inhibitor is working. The inclusion of a positive control such as injected beads that trigger the melanization reaction by the immune system in the presence and absence of PTU would make the argument more convincing. In addition, thes PTU treatment seems to have a significant negative impact on the survival of the flies. It would be more convincing if the authors dissected out tissues from PTU and NTBC treated flies in a similar manner to what is shown in Figure 2 A+B to show that the melanization found in the NTBC treated flies alone disappears in the presence of PTU. Another alternative would be for the authors to mine their metabolomics data to compare the levels of downstream metabolites of the PO cascade such as dopa, dopamine and melanin in the NTBC treated flies.

The authors state that both oral and topical administration of NTBC are effective delivery mechanisms and that dietary protein is essential for its toxicity. In Supplemental figure 8 they show that the LC50 of NTBC requires ingestion of ~15 mg/ml of protein. This experiment could also use a control group which was fed BSA in the absence of NTBC. Blood is a complex mixture which tsetse have evolved specific mechanisms to deal with, however ingestion of sugar solution infused with BSA has a very different composition/osmolarity to that of blood and may have significant biochemical impacts on the fly. The control group would allow such issues to be accounted for and isolate the effect of protein concentration on NTBC mode of action.

As the headline of one of the major sections of the results indicates that "NTBC-associated damage does not depend on phenoloxidase activation or haem, but protein concentration in the bloodmeal" I think either this section should be toned down or additional controls should be added to the experiments to bolster this statement.

In the final section which discusses the results of the mathematical modelling of tsetse populations a model is presented which predicts how long NTBC treatment interventions would be required to reduce the population replacement number below 1. I was somewhat confused as to how to interpret the color scale as it would seem that the lower the number of days of treatment required would represent higher efficacy. However, the black indicator at the bottom of the figure indicates that the population is not controlled. The color indicating only 10 days of treatment at the bottom of the gradient is very close to black, so to my eye treatment efficacy is correlating with color intensity with the exception of completely black dots which are ineffective. If I am interpreting this correctly, I would suggest to reverse the gradient or change the color of the dots that fall into the "Not controlled" category. Another consideration in regards to this figure is that the two panels represent treatment of humans alone and treatment of livestock and humans. Treatment of healthy humans with a drug primarily tested in people with a genetic disease could potentially have unanticipated long term side effects. I think consideration should be given to treatment of livestock alone and that that should be presented as a third condition.

Overall this is a very nice paper with exciting potential that requires some additional tweaking prior to publication.

Reviewer #3: The manuscript by M. Sterkel and colleagues describes the effects of genetic and chemical impairment of tyrosine catabolism in two species of Glossinia, the tsetse fly vector of both human and animal African trypanosomiasis, a neglected tropical disease. The authors demonstrate that perturbation of tyrosine catabolism, either through mRNA silencing targeted against tyrosine aminotransferase (TAT) and hydroxyphenylpyruvate dioxygenase (HPPD), or through the use of the selective HPPD inhibitors mesotrione and nitisinone, causes widespread tissue damage, morbidity and mortality in Glossinia spp. following a vertebrate blood meal, or digestion of protein. The authors go on to characterize this effect in detail, demonstrating the insecticidal effects of HPPD inhibitors when included in sugar or blood meals, or through direct external application. The authors conclude that HPPD inhibitors could be used as endectocides in mass drug administration (MDA), livestock treatment, or as an active ingredient in tsetse-targeting interventions. New tools to control tsetse flies are needed, and the presented data are an encouraging step forward. 

The authors should be commended for presenting a large amount of work on an important subject. However, there are a number of important issues that reduce the relevance of this study.

MAJOR ISSUES

This study follows a previous publication from some of the same authors detailing similar effects of TAT and HPPD RNAi, and mesotrione and nitisinone on the Trypanosoma cruzi vector Rhodnius prolixus, and the mosquito vector Aedes aegypti. While the present manuscript goes into greater detail on the effects of perturbation of tyrosine catabolism in tsetse, the prior, similar findings in other hematophagic insects strongly reduces the novelty and impact of the present work.

Throughout, revisions are needed in the presentation of the data. The authors leave the reader to do much of the work to synthesize the data as the results section is, in general, too vague throughout (see specific comments). Data, including dosage concentrations, diluents, controls, median time-to-death, percent mortality at 24 h, should be presented clearly in the results section as well as the method text and not left to the figures. The same is true for figure legends, which lack detail. 

Figures are difficult to follow, with very similar looking panels presenting different treatment conditions. All figures should have treatment conditions specified either on top of the graphs or on the x axis.

Line 195 Although the addition of phenylthiourea may indicate that PO activation is not required for NTBC activity, those experiments do not prove this is the case as the authors do not determine whether PO activity was inhibited in those conditions. So those conclusions aren't warranted. 

Line278 No details of the model used are provided in the results. Also, the description of those results is minimal. What's the possible impact of the two control measures, and at what coverage level is an impact achieved? 

The authors should also discuss the feasibility of achieving such high coverage levels as those indicated by the model figure.

'Data not shown' should be presented in supplementary figures.

MINOR ISSUES

Results

The use of "spiked" to describe solutions containing treatment compounds should be avoided. Instead include total makeup of provided solution (e.g. defibrinated horse blood with 10 µM mesotrione). In general, the range, or specific concentrations used in experiments should be stated in the results.

Line 102-104: state degree of mortality, and knockdown efficiency, for each treatment. Presumably the explanation for incomplete mortality is due to incomplete knockdown, but this is not stated in the text.

Line 121: please provide specific safety information for nitisinone/NTBC. 

Related: nitisinone should be defined once as NTBC at first use, then referred to as that only for the remainder of the text. Currently, references to this drug are inconsistent. 

Line 126: missing reference for standard oral dose of 1 mg/kg nitisinone. 

Line 129 to 131: There is an extensive literature on both veterinary and human endectocides that should be referenced here. 

Line 132-134: These data should be described in more detail. 

Line 136-143: Again the data should be described fully - median-time-to-death etc - including the fact that mortality was incomplete, and (as stated in the discussion, but not here) that surviving flies suffered some morbidity effects

Line 155-159: The text (and figure legends) for figure 2 describe features that are not readily apparent on the included micrographs. Please highlight specific features of interest on figures. 

Line 160-173: as above

Line 200-208: It is not clear whether these data were generated with mesotrione or nitisinone. The section heading states NTBC, but supp. Fig. 7 states mesotrione. 

Line 209-215: state concentration of NTBC, state concentration range of BSA, concentration of sugar (fructose in this case). 

Line 217-225: State method of application, vehicle, volume, concentration range applied. See comment above on topical application. 

Line 218: Provide exact numbers of flies on the graph. Specify which sex was tested, and which line corresponds to which concentration.

Line 240: it's not clear why the authors used p450s from mosquitoes rather than from tsetse. This limits the relevance of these findings.

Line 243: define "moderately"

Line 267: sugar solutions of NTBC didn't kill tsetse flies either. The authors should test topical application in bees, as possible mortality effects in bees may be unrelated to protein uptake. Moreover, the conclusions that NTBC selectively kills blood-sucking insects provided in the discussion (line 386) is not justified given the limited number of insect species tested here and in previous work.

Discussion 

Line 347: what is the approximate tyrosine content in whole blood (bovine, human). The comparison is interesting. 

Caution is to be encouraged in discussing the unknown environmental safety of NTBC which, given its origin as an herbicide (albeit an agriculturally unsuitable one) may have some environmental impact. See also comment above about beneficial insects.

The suggestion that nitisinone could be used in a human MDA approach is presented without scrutiny. Given the potential ethical issues of administering a human therapeutic en masse with no direct benefit to the individual, I would like to see some discussion of these issues.

Topical application of compounds dissolved in a volatile vehicle (e.g. acetone), should not be considered a direct proxy for compound contact on a flag-trap, surface-treated animal, or similar substrate. Acetone disrupts the waxes of the insect exocuticle and, as a result, facilitates traversal of dissolved compounds into the insect hemocoel. While the fact that external exposure to nitisinone is effective is an important finding, some discussion of the limitations of this technique should be included.

Reviewer #4: I am only reviewing the modeling portion of this paper.

The authors use a simple model to estimate the impact of distributing NTBC as an endectocide for HAT, to humans and/or livestock, and predict the dosing interval for NTBC needed to drive R below 1. It's great to see modeling deployed to estimate the potential impact of a new tool like this! However, I have a few concerns with model structure and additional concerns about parameterization:

In the model, infection in animals is the same as infection in humans. The model says animal infections are just as transmissible and it is not at all clear that this is the case for T.b. gambiense, which accounts for the vast majority of HAT infections.

Why exp(-\\lambda t^2) instead of exp(-\\lambda t) for the drug decay?

Equation for m_2 suggests effective coverage of 100% for humans (or animals) treated with NTBC. This is unrealistically high. Operationally realistic excellent coverage with mass drug administration, which this is, would probably not exceed even 80%.

No evidence that model can capture observed HAT dynamics: no description of model calibration to HAT incidence data or other model validation. Would be great to see this in a revision so that we can be confident that the model is capturing relevant transmission. Current control methods also rely heavily on treatment, which in recent years has become much less onerous. It's probably fine to ignore treatment in a paper that is very focused on a novel vector control method but it should be acknowledged that no one would pursue mass NTBC distribution unless good case management were already in place.

The authors seem unfamiliar with the existing HAT modeling literature from a number of groups, including Kat Rock and Nakul Chitnis, for example:

Kat S Rock, Martial L Ndeffo-Mbah, Soledad Castaño, Cody Palmer, Abhishek Pandey, Katherine E Atkins, Joseph M Ndung'u, T Déirdre Hollingsworth, Alison Galvani, Caitlin Bever, Nakul Chitnis, Matt J Keeling, Assessing Strategies Against Gambiense Sleeping Sickness Through Mathematical Modeling, Clinical Infectious Diseases, Volume 66, Issue suppl_4, 15 June 2018, Pages S286-S292, 

Stone CM, Chitnis N. Implications of heterogeneous biting exposure and animal hosts on Trypanosomiasis brucei gambiense transmission and control. PLoS Comput Biol. 2015 Oct 1;11(10):e1004514.

---

## [Decision Letter · Decision Letter 2]

13 Oct 2020

Dear Alvaro,

Thank you very much for submitting a revised version of your manuscript "Repurposing the orphan drug nitisinone to control the transmission of African trypanosomiasis" for consideration as a Research Article at PLOS Biology. This revised version of your manuscript has been evaluated by the PLOS Biology editors, the Academic Editor and three of the original reviewers (2, 3 and 4).

All reviewers appreciate the work performed during revision and the article is almost ready, although as you will see reviewers 3 and 4 have some remaining issues that will need to be addressed in a revision that we anticipate should be straightforward. We will assess your revised manuscript and your response to the reviewers' comments in-house (and with our Academic Editor) and may need to run your revision quickly by reviewer 4. We expect to receive your revised manuscript within 1 month, please let us know if this is likely to take longer.

I have also gone through your work in detail and have noticed the following points that will need addressing:

- The control BSA group figure that you provided in the response to reviewers should be included in the manuscript, as readers may have the same concern

- The manuscript should reference the Oliveira lab preprint (bioRxiv doi.org/10.1101/669747) on the use of HPPD inhibitors for mosquito control that you allude to in response to reviewer 1 and discuss the implications of those results (it increases the relevance of the work to say if the strategy could be used for mosquitoes control, given their importance as vectors of disease)

- Your metabolomics dataset should be deposited in an appropriate public repository, such as Metabolights, and the accession number included in the Data Availability statement (in the online article submission form) and were relevant in the manuscript. 

- Figure 2A needs a scale bar 

- Units need to be provided for scale bar in the legend to SI 1

- Statistical information (n, whether the mean or median is plotted, what kind of error bars, etc) is missing in the legend of figure 1, for the box-and-whisker plot data in 3B , and for Suppl Figure 4

- As there are no length restrictions, please include all of the Material and Methods information in the main text. The detailed explanation of the mathematical model can be left in SI, and the file labelled accordingly, but the other two subheadings should be moved to the main text.

Note that all individual quantitative observations that underlie the data summarized in the main and supplementary figures of your paper need to be made available as source data. Each figure legend should include a statement indicating where the numerical source data for the figures can be found (e.g. Numerical source data underlying this figure can be found in XXX). You can provide source data in one of the following forms:

2) Deposition in a publicly available repository such a GitHub. If you choose this option, please also provide the accession code or a reviewer link so that we may view your data before publication.

Regardless of the method selected, please ensure that you provide the individual numerical values that underlie the summary data displayed in all of the main and supplementary figures that present graphs, as they are essential for readers to assess your analysis and to reproduce it:

Please also ensure that figure legends in your manuscript include information on where the underlying data can be found, and ensure your supplemental data file/s has a legend that makes it easy to understand.

- The Supplementary Figures need to be provided as individual files

-----

**IMPORTANT - SUBMITTING YOUR REVISION**

*Resubmission Checklist*

*Published Peer Review*

With kind regards,

Nonia

Nonia Pariente, PhD,

Editor-in-Chief,

npariente@plos.org,

PLOS Biology

REVIEWS:

Reviewer expertise:

Reviewer #2: tsetse physiology and trypanosomiasis control

Reviewer #3: strategies to prevent malaria transmission; mosquito vector control; malaria pathogenesis

Reviewer #4: mathematical modelling of malaria transmission and effects of antimalarial strategies

Comments to authors:

Reviewer #2: Thank you to the authors for their comprehensive response to my and the other reviewers critiques. The manuscript is much improved and the issues raised in my previous review have been addressed.

Reviewer #3: This is a greatly improved resubmission. The authors have done a good job at addressing all comments, adding more information when needed and more comprehensively explaining their findings, as well as their significance and limitations. I just have some minor revisions to add:

Line 320: provide references for this statement 

Line 391: I think the word 'albeit' doesn't fit here

Lines 408-430. I particularly appreciated the more detailed discussion of possible emergence of tsetse resistance. However, insects are capable of evolving multiple resistance mechanisms besides target site and metabolic resistance, and I encourage the authors to acknowledge that eventually flies would develop resistance to their compound if it was used as monotherapy, as they suggest in line 438. 

Reviewer #4: Two main points on the modeling section:

1. Clarity on model assumptions

2. Relevance of NTBC to public health

I get what the authors are saying re: model validation. Their goal is to use the simple Ross-Macdonald and see what happens if mortality of the vector is increased through endectocide. But the authors need to do a much better job being explicit about the assumptions of their model because the modeling section of the paper is key to showing that that NTBC has relevance for public health. 

It's somewhat unclear to me how the Re values with intervention are being calculated. Is m2(t) plugged into the analytical solution to R to derive an Re(t)? Then are the Re values shown in Fig 5 a mean Re? Or is Re(t) being estimated directly from the simulation with a method like EpiEstim? I'm a little bit confused by the supplemental methods description.

Either way, does this mean that the modeling assumes that NTBC distribution continues indefinitely? I think this is a key assumption that should be explicitly mentioned to help readers interpret the modeling results. Furthermore, there's a key assumption of R0 = 1.1 that is not stated explicitly in the main text. 

My read of Fig 5 and Fig S13 is that NTBC in its current form is not an operationally relevant potential vector control: monthly MDAs (or even 3-monthly) at 80% coverage aren't going to be sustainable for long enough to interrupt transmission, and the modest dip of Re to just below 1 in most of the searched parameter space would indicate that a long program of MDAs would be needed. Given the low case numbers of both rhodesiense and gambiense HAT, WHO is focusing on improved diagnostics and treatments to interrupt transmission (gambiense) or eliminate as a public health problem (rhodesiense). The modeling results for NTBC don't suggest that it would drive a shift in HAT policy toward vector control.

---

## [Editor Report · Decision Letter 3]

9 Nov 2020

Dear Alvaro,

Thank you for submitting your revised manuscript entitled "Repurposing the orphan drug nitisinone to control the transmission of African trypanosomiasis" for publication in PLOS Biology. I have now discussed the revision with our Academic Editor and I am happy to contact you with an accept in principle decision, conditional on a few last changes to comply with our reporting and formatting requirements for publication.

- The Data Accessibility Statement that you fill in during submission in our online system will be published with your paper. Please make sure that it is complete and includes, e.g. information about the Metabolights deposition and the accession code.

- Before final sign off we will need to access the Metabolights data. As it is not yet live, please confirm that it is set to live on publication and provides us with a temporary access token to look at it before formal acceptance.

- A member of our team will be in touch shortly with an additional set of requests. As we can't proceed until all have been addressed, your swift response will help prevent delays to publication.

- a cover letter that should detail your responses to any editorial requests, if applicable

*Copyediting*

*Published Peer Review History*

*Early Version*

We expect to receive your revised manuscript within two weeks. Please do not hesitate to contact me should you have any questions.

With best wishes,

Nonia

Nonia Pariente, PhD,

Editor-in-Chief,

npariente@plos.org,

PLOS Biology

Reviewer remarks:

---

## [Editor Report · Decision Letter 4]

30 Nov 2020

Dear Dr Acosta-Serrano,

On behalf of my colleagues and the Academic Editor, Nora J Besansky, I am pleased to inform you that we will be delighted to publish your Research Article in PLOS Biology. 

PRODUCTION PROCESS

Before publication you will see the copyedited word document (within 5 business days) and a PDF proof shortly after that. The copyeditor will be in touch shortly before sending you the copyedited Word document. We will make some revisions at copyediting stage to conform to our general style, and for clarification. When you receive this version you should check and revise it very carefully, including figures, tables, references, and supporting information, because corrections at the next stage (proofs) will be strictly limited to (1) errors in author names or affiliations, (2) errors of scientific fact that would cause misunderstandings to readers, and (3) printer's (introduced) errors. Please return the copyedited file within 2 business days in order to ensure timely delivery of the PDF proof. 

If you are likely to be away when either this document or the proof is sent, please ensure we have contact information of a second person, as we will need you to respond quickly at each point. Given the disruptions resulting from the ongoing COVID-19 pandemic, there may be delays in the production process. We apologise in advance for any inconvenience caused and will do our best to minimize impact as far as possible.

EARLY VERSION

PRESS 

Kind regards,

Erin O'Loughlin

Publishing Editor, 

PLOS Biology

on behalf of

Nonia Pariente, PhD,

Editor-in-Chief

PLOS Biology